# Satellite observations for describing fire patterns and climate-related fire drivers in the Brazilian savannas

Guilherme Augusto Verola Mataveli[1], Maria Elisa Siqueira Silva[1], Gabriel Pereira[1,2], Francielle da Silva Cardozo[2], Fernando Shinji Kawakubo[1], Gabriel Bertani[3], Julio Cezar Costa[2], Raquel de Cássia Ramos[2], and Viviane Valéria da Silva[2]

[1]Department of Geography, University of São Paulo, São Paulo, 05508-000, Brazil
[2]Department of Geosciences, Federal University of São João del-Rei, São João del-Rei, 36307-352, Brazil
[3]Remote Sensing Division, National Institute for Space Research, São José dos Campos,12227-010, Brazil

*Correspondence to*: Guilherme Augusto Verola Mataveli (mataveli@usp.br)

**Abstract.** In the Brazilian savannas (Cerrado biome) fires are natural and a tool for shifting land-use, therefore, temporal and spatial patterns result from climate, vegetation condition and human activities interaction. Moreover, orbital sensors are the most effective approach to establish patterns in the biome. We aimed to characterize fire, precipitation and vegetation condition regimes and to establish spatial patterns of fire occurrence and their correlation with precipitation and vegetation condition in the Cerrado. The Cerrado was, respectively, second and first biome for the occurrence of hotspots and burned area (BA). Occurrences are higher during the dry season and in the savanna land-use. Hotspots/BA are tending to decrease, and concentrate in the Northern, however, more intense hotspots are not necessarily located where concentration is higher. Spatial analysis showed that averaged/summed values can hide patterns, such as for precipitation, which have lowest average in August, however, minimum precipitation in August was found in 7 % of the Cerrado. Usually, there is a 2-3 months lag between minimum precipitation and maximum hotspots/BA, while minimum VCI and maximum hotspots/BA occur in the same month. Hotspots/BA are better correlated with VCI than precipitation, qualifying VCI as an indicator of the susceptibility of vegetation to ignition.

## 1 Introduction

A common process in most of the land areas of the world (Hantson et al., 2013), fires consume large areas of vegetation across the Earth's surface, modifying its characteristics (Vadrevu et al., 2014) and playing an important role in climate due to the emissions related to biomass burning (Kaiser et al., 2012). Moreover, extensive fire activity disturbs the ecosystems, decreasing plants and animal species and causing soil depletion (Fearnside, 2000), and causes social and economic costs (Brunson and Tanaka, 2011; Stephenson et al., 2013).

The frequency of fire weather seasons is becoming longer, once between 1979 and 2013 fire weather seasons have lengthened in more than 25% of the world's vegetated area, increasing in about 19% the global mean fire weather season length (Jolly et

al., 2015). Globally, around 25 % of the areas which have a significant fire activity are characterized by a bimodal pattern, presenting at least one fire season occurring under unfavorable fire weather conditions, which indicates an anthropic influence in the occurrence of fires; predominantly bimodal pattern areas are mostly located in Canada-Northern USA, Central Eurasia, Eastern Siberia, Northern India and Southern South America, highlighting the importance of population density and land

management techniques in the global fire regime (Benali et al., 2017). In addition to anthropic action, climate is a powerful driver for the occurrence of fires, able to control the amount of fuel available for burning and enhancing the probability of ignition. While higher amount of precipitation during the dry season reduces the occurrence of fires, rainier years during the dry season can increase fire activity in the following year (Archibald et al., 2010). Moreover, studies of long-term trends and inter-annual variability of fires found large year-to-year changes associated with extreme climate conditions (Chen et al.,

2013). Combined with climate, fuel moisture and fuel type, which are characteristics of the vegetation, are dependent variables for the ignition and spread of fires (Leblon et al., 2012). Variations in vegetation condition, which can be expressed by vegetation indices, are an important indicator of stress factors for plants, such as drought, and are useful for assessing the susceptibility of vegetation to fires (Bajocco et al., 2015). Studies have already shown the correlation between the seasonal variability of fuel moisture and fire occurrence, as well as the intra-annual and inter-annual variability of vegetation condition

due to shorter or longer dry seasons (Chéret and Denux, 2011). Therefore, the interaction of human activities, climate and vegetation define the temporal and spatial patterns for the occurrence of fires, highlighting the importance of characterizing the spatial patterns of fire occurrence and their correlation with climatic variables and vegetation condition, which is necessary to understand the dynamics of areas affected by fires, especially in areas suffering severe changes in land-use and land-cover (LULC) and presenting a climatic seasonality, such as the Cerrado biome (Brazilian savannas).

According to Archibald et al. (2010), the relationship between climate controllers and fire occurrence is complex and varies for each biome. The tropical savannas are the most frequently burned ecosystems in the world (van der Werf et al., 2008; Bowman et al., 2009), where humans are responsible for most of fires (Archibald et al., 2012), especially used for opening areas for agriculture and livestock farming, and for pest control (Shimabukuro et al., 2013). In the Cerrado, the second largest biome in South America and mostly constituted of short grassland vegetation, natural fires are common due to lightning

(Ramos-Neto and Pivello, 2000). Moreover, since the 1970s this biome is suffering an intense agricultural expansion process (Fearnside, 2001), being fire an important tool in this land-use and land-cover change (LULCC) process for removing the natural remnants of vegetation, as well as shifting cultivations, eliminating crop residues or stimulating the regrowth of herbaceous plant for cattle feeding during the dry season (Pivello, 2011). These reasons qualify the Cerrado as an important biome to be studied, however, according to Beuchle et al. (2015), despite the increasing anthropic pressure in the Cerrado,

LULCC in the biome has been overlooked until recently when compared to the efforts for monitoring LULCC in the Amazon biome. The absence of an effective fire policy also contributes to the LULCC process in the natural remnants of Cerrado, which is the most biodiverse savanna in the world (Durigan and Ratter, 2016).

Regarding the Cerrado, we can highlight studies that analysed fires and precipitation, such as Moreira de Araújo et al. (2012), Libonati et al. (2015a) and Moreira de Araújo and Ferreira (2015), which used averaged/summed values of burned area,

hotspots or precipitation for the entire Cerrado, a biome presenting an area of more than 2 million km$^2$, therefore, the variation of these variables within the biome was not analysed. Still, Libonati et al. (2015b) and Libonati et al. (2016) analysed the spatial and temporal distribution of burned area in the Cerrado, however, these works did not establish the spatial distribution of correlation between burned area/hotspots and precipitation or vegetation condition, which is an important approach for

understanding the complex climate-vegetation-fire occurrence relationship. It should also be mentioned the recent study of Nogueira et al. (2017), which analysed the sensitivity of fire danger indices (FDIs) in describing the seasonal fire danger in the Brazilian biomes, including the Cerrado, by correlating distinct FDIs with burned area datasets, and found that the sensitivity of FDIs is biome-specific.

Considering the phenomenon analysed, data derived from orbital sensors is the most efficient source of information to

comprehend the fire dynamics, once it allows to observe large areas of the surface daily and repeatedly (Ichoku et al., 2012; Hantson et al., 2013; Andela et al., 2016). Among the orbital sensors available for studying fires, Moderate Resolution Imaging Spectroradiometer (MODIS) active fire and burned area products were the first data sets derived from the new generation of moderate resolution orbital sensors (Giglio et al., 2016) and have been widely used to study fires in the tropical savannas of Brazil (Nascimento et al., 2011; Moreira de Araújo et al., 2012; Libonati et al., 2015a; Libonati et al., 2015b; Moreira de

Araújo and Ferreira, 2015; Libonati et al., 2016), Africa (Archibald et al., 2010; Kusangaya and Sithole, 2015), Australia (Andersen et al., 2005; Yates et al., 2008; Maier et al., 2013), and in the entire tropical savannas (van der Werf et al., 2008). It should also be mentioned the use of MODIS data for estimating vegetation condition from vegetation indices (Chéret and Denux, 2007; Chéret and Denux, 2011), while precipitation is mostly estimated using Tropical Rainfall Measurement Mission (TRMM) satellite data (Kummerow et al., 2000), once it provides good estimates of the spatial and temporal patterns of

precipitation and is widely validated.

Given the information presented above, we aimed to characterize the fire, precipitation and vegetation condition regimes and to use spatial analysis tools to establish spatial patterns of fire occurrence and their correlation with precipitation and vegetation condition in the Cerrado using data obtained from orbital remote sensing during the 2002-2015 period.

## 2 Materials and Methods

### 2.1 Study Area

The Cerrado (Fig. 1) is the second largest Brazilian biome, only smaller than the Amazon biome, and covers approximately 22 % of the Brazilian territory in 11 different states (IBGE, 2010). The biome is characterized by a wide climatic variability, once it covers a large area extended for many latitudinal belts (Kayano and Andreoli, 2009), usually presenting a dry season from May to September and a rainy season from October to April for most of the Cerrado (Coutinho, 1990; Pivello, 2011;

Moreira de Araújo et al., 2012). Due to the high biological diversity, especially endemic species that represent approximately 44 % of the flora (Klink and Machado, 2005), the Cerrado is a world biodiversity hotspot (Myers et al., 2000). Among the factors that explain the biodiversity in the biome, we can highlight a strong climatic seasonality, water availability and

anthropic disturbances, such as deforestation and fires (Coutinho, 1990). The Cerrado is composed of different vegetal formations, presenting: I) Grasslands, mainly constituted by herbaceous species and certain shrubs without the presence of tree species; II) Savannas, presenting sparse tree and shrubs over a grassy extract; and III) Forest formations, where tree species with continuous or discontinuous canopy dominate (Dias, 1992).

However, the LULCC process caused by the agricultural expansion since the 1970s (Fearnside, 2001) significantly reduced the natural vegetation of the biome. Before the 1970s, the Cerrado was considered unsuitable for agriculture due to the poor soils in the biome, although, the advances in agricultural techniques, favorable conditions for mechanization, government incentives and the low price of land contributed to transform the Cerrado into a growing agricultural region (Bickel and Dros, 2003). The anthropic fires in the natural remnants of the Cerrado, caused by the LULCC process, are commonly performed

during the dry season, opposed to the natural fires, which usually occur in the beginning of the rainy season, and are favouring the predominance of herbaceous species or causing land degradation in the remnants of the Cerrado (Pivello, 2011).

## 2.2 MODIS fire products

MODIS sensor, designed for studying the dynamics of the biosphere, is onboard the polar orbit Terra and Aqua satellites, providing data in 36 spectral channels between 0.4 μm and 14.4 μm with nominal spatial resolution ranging from 0.25 to 10

km, depending on the product, temporal resolution of 1 to 2 days and enabling four daily imaging surveys from the same surface (Justice et al., 2002; Giglio et al., 2003). MODIS active fire products (MOD14 and MYD14 products, derived from the MODIS sensor onboard Terra and Aqua satellites, respectively) detect burning pixels with nominal spatial resolution of 1 km using a contextual algorithm that applies thresholds to the middle−infrared and thermal infrared brightness temperature (T4μm and T11μm, respectively) (Giglio et al., 2003). Still according to Giglio et al. (2003), MODIS has two 4 μm channels

(21 and 22), and, usually, channel 22 is used to detect active fires, however, when it saturates or has missing data, it is replaced with channel 21, while T11μm is derived from channel 31; false detections are rejected by examining the brightness temperature relative to adjacent pixels. In addition to the spatial location, MODIS active fire products also provide other information regarding hotspots detected, such as Fire Radiative Power (FRP), which is defined as the rate that energy is emitted as electromagnetic radiation during the combustion process and is related to the intensity of fires and biomass burning (Wooster

et al., 2005).

Regarding burned area, the MODIS globally gridded 0.5 km burned area product (MCD45A1) (Roy et al., 2005) contains burning and quality information on a per-pixel basis monthly. MCD45A1 algorithm analyses the daily surface reflectance dynamics from both MODIS sensors for detecting the approximate date of burning and mapping the spatial extent of recent fires, enabling to recognize spatial patterns of burned area (MODIS Fire Products ATBD, 2006).

## 2.3 TRMM precipitation data

TRMM satellite results from a joint program between the United States of America and Japan space agencies and aims to provide precipitation data in the tropical and subtropical areas of the globe (Kummerow et al., 2000). TRMM satellite is

operational since 1997 and equipped with three different sensors: I) Precipitation Radar (PR), built to provide a 3-D view of rainfall distribution over the tropics and subtropics; II) Microwave Imager (TMI), which aims to analyse the content of the integrated precipitation column, cloud liquid water and ice, and rain intensity and type; and III) Visible and Infrared Scanner (VIRS), a sensor designed for observing clouds type, coverage and top temperature (Kummerow et al., 1998). For analysing the precipitation regime in the Cerrado biome between 2002 and 2015, TRMM monthly precipitation product (3B43), provided with spatial resolution of 0.25° (approximately 30 km) in millimeters per month (mm month$^{-1}$), was used. TRMM satellite monthly estimates have been validated for Brazil by Pereira et al. (2013), who compared TRMM data with monthly precipitation estimated by 183 meteorological stations during the 1998-2010 period and found strong correlation between TRMM data and the meteorological stations in Brazil, however, TRMM tends to overestimate monthly precipitation in 15 %.

## 2.4 MOD13A3 product and Vegetation Condition Index (VCI)

The use of vegetation indices is an important approach for monitoring both vegetation and LULC classes. Considering the goals of MODIS sensors, global vegetation indices derived from MODIS provide consistent spatial and temporal comparisons of vegetation conditions (Justice et al., 2002). The global MODIS 1 km Normalized Difference Vegetation Index (NDVI-MOD13A3 product, based on MODIS/Terra data) is monthly provided and considers all data from the 16 days 1 km products that overlap the month employing a weighted temporal average if data is cloud free, or a maximum value in case of clouds (MODIS Vegetation Index ATBD, 1999; Didan, 2015). In this study, MOD13A3 product was used to estimate the Vegetation Condition Index (VCI) in the Cerrado biome during the period between 2002 and 2015.

Satellite-based drought indices, such as the NDVI based VCI (Kogan, 1995), are important sources of information for detecting the occurrence, the duration, the intensity and the impacts of drought (Quiring and Ganesh, 2010; Jiao et al., 2016). Initially proposed to evaluate the global weather impact over vegetation and later proved useful for evaluating wildfire danger, VCI is a relative measurement of the NDVI value at the observation date with respect to extreme conditions of NDVI over a reference period (Chéret and Denux, 2007), as shown in Eq. (1):

$$VCI = \frac{NDVI - NDVI_{Min}}{NDVI_{Max} - NDVI_{Min}} * 100 \tag{1}$$

where NDVI, $NDVI_{Min}$ and $NDVI_{Max}$ correspond to the NDVI, NDVI minimum and NDVI maximum values, respectively. According to Chéret and Denux (2007), one of the most influent variables for assessing fire danger is the vegetation condition, being remote sensing capable of providing relevant information about this variable and being VCI able to represent this variable. Thenkabail et al. (2004), who analysed different vegetation indices for monitoring drought and vegetation condition in Southwest Asia, also proved VCI as a sensitive indicator for monitoring vegetation condition, presenting better results than other indices, such as the Temperature Condition Index (TCI). The great advantages of using VCI are that it can be easily estimated, it does not require station observation data and it can provide near real-time estimates over the globe at a relatively high spatial resolution (Quiring and Papakryiakou, 2003).

**2.5 MODIS Land Cover Type Product (MCD12Q1)**

The MCD12Q1 product provides global LULC maps annually with spatial resolution of 500 meters using 8-days composite data from both MODIS sensors (Friedl et al., 2010). LULC classes are determined from a supervised classification algorithm based on high quality land-cover training sites, also including as specific inputs Normalized BRDF-Adjusted Reflectance and land surface temperature derived from MODIS data, and are provided in five distinct land-cover classification systems (MODIS Land Cover Product User Guide, 2013). In this work, we have used MCD12Q1 data based on the International Geosphere-Biosphere Programme (IGBP) classification system, which divides global LULC into 17 distinct LULC classes and is described in Friedl et al. (2002). The validation study of the MCD12Q1 product performed by Friedl et al. (2010) showed an overall accuracy of 75 %, despite the range in class-specific accuracies.

**2.6 Data Processing**

Initially, all hotspots detected by the collection 6 MODIS active fire products in the Brazilian territory between 2002 and 2015 were grouped according to the spatial delimitation of the Brazilian biomes proposed by IBGE (2010), aiming to analyse the contribution of the hotspots occurred in the Cerrado nationally. Sequentially, the time series of monthly total hotspots in the entire Cerrado biome was generated considering the date of the occurrence of the hotspots available in the MODIS active fire products. Still, total hotspots for each year analysed and monthly average of hotspots were calculated. Aiming to analyse the major land-uses related to the occurrence of fires in the biome, hotspots detected by the MODIS active fire products were crossed with the annual LULC maps derived from collection 5.1 MCD12Q1 product according to the spatial location of the hotspots. Firstly, all annual MCD12Q1 LULC maps following the IGBP classification system were clipped to the delimitation of the Cerrado proposed by IBGE (2010) and reclassified to seven LULC classes: savannas, woody savannas, grasslands, croplands, cropland/natural vegetation mosaic, evergreen broadleaf forest (LULC classes that compose around 98 % of the Cerrado, as will show below) and other land-uses. It should be mentioned that the most recent annual LULC map derived from the MCD12Q1 product was produced for the year 2013, therefore, the spatial location of the hotspots detected in the Cerrado during the years 2014 and 2015 was crossed with the 2013 LULC map. The total and percentage of hotspots in the reclassified land-uses of the Cerrado for the entire 2002-2015 period was then calculated, as well as the area of each reclassified land-use during the period analysed.

Regarding burned area analysis, initially all the 16 tiles of the collection 5 MCD45A1 product which cover the Brazilian territory were acquired and monthly burned area pixels were grouped and summed according to the spatial delimitation of the Brazilian biomes, also aiming to evaluate the contribution of burned area occurred in the Cerrado nationally, as proposed for the hotspots. Then, only pixels flagged as one, which are highly reliable observations, and, therefore, the most probable pixels of being burned area, were considered, as previously proposed by Moreira de Araújo et al. (2012) and Moreira de Araújo and Ferreira (2015) for studying burned area in the Brazilian biomes and in the Cerrado, respectively. For the analysis of the burned area occurring only in the Cerrado biome, the 5 tiles covering the delimitation of the Cerrado biome were considered and

processed as described above. Finally, the time series of monthly total burned area in the entire Cerrado biome was generated for the 2002-2015 period, as well as annual estimates of burned area for each year analysed and monthly average burned area. For TRMM data processing, version 7 3B43 product monthly images were obtained for the entire tropical and subtropical area of the globe and clipped to the study area according to the delimitation of the Cerrado proposed by IBGE (2010). Then, monthly

and annual average precipitation for the entire Cerrado biome during the period between 2002 and 2015 were estimated. VCI was estimated using the global 1 km monthly NDVI MODIS product. Initially, all the steps described for MCD45A1 in the Cerrado were performed for the collection 6 MOD13A3 product. Sequentially, maximum and minimum NDVI for each pixel of the study area were estimated based on MODIS NDVI time series from 2002 to 2015, which is necessary to define the maximum and minimum NDVI masks used to estimate VCI. Finally, monthly and annual VCI for the entire Cerrado biome

were estimated using Eq. (1); for annual VCI, the term NDVI in Eq. (1) corresponded to the annual average NDVI of each pixel located in the study area.

Boxplot analysis for the Cerrado time series of monthly total hotspots, monthly total burned area, monthly average precipitation and monthly average VCI was performed in order to analyse the variation of these variables during the months and years studied and to identify outliers. Moreover, Breaks For Additive Seasonal and Trend (BFAST), an additive method that

decomposes a time series into seasonal, trend and noise components (Verbesselt et al., 2010), was applied to the Cerrado time series of monthly total hotspots, monthly total burned area, monthly average precipitation and monthly average VCI aiming to find trends in the four time series analysed. According to Verbesselt et al. (2010), BFAST assumes that the trend component ($T_t$) is piecewise linear with break points, here, as well as the maximum number of interactions, considered as 1. The significance of the trend component found using BFAST for the four variables analysed was tested using the t-Student test

with significance level of 5 %.

Regarding the spatial analysis, all maps described here and presented in the results considered the same grid size (0.25º x 0.25º, spatial resolution of TRMM data), so all results are comparable. The spatial distribution of the hotspots within the Cerrado was analysed by summing all the hotspots occurring in the Cerrado during the 2002-2015 period in each cell of the grid size described above. Additionally, the spatial pattern of FRP estimated by the MODIS active fires in the Cerrado was analysed by

calculating the mean FRP of all hotspots detected in each grid cell of the biome considering the entire 2002-2015 period.

The spatial analysis also estimated the month with highest incidence of hotspots and burned area, minimum amount of precipitation and minimum VCI in the Cerrado considering the 168 months analysed. Minimum or maximum monthly incidence was established according to the average of the sum of monthly hotspots, sum of monthly burned area, monthly average precipitation and monthly average VCI for each of the twelve months in each grid cell of the Cerrado. Furthermore,

the spatial monthly lag between the minimum of precipitation and maximum of hotspots, minimum of precipitation and maximum of burned area, minimum of VCI and maximum of hotspots and minimum of VCI and maximum of burned area was also calculated.

Spatial statistical analysis consisted on calculating the Pearson's Correlation Coefficient (R), which shows the linear relationship between two datasets, for each grid cell of the regular grid over the Cerrado considering the month by month

values during the 168 months comprehended between the 2002-2015 period for four pairs of variables: monthly total hotspots and monthly average precipitation, monthly total burned area and monthly average precipitation, monthly total hotspots and monthly average VCI and between monthly total burned area and monthly average VCI. In here, monthly total hotspots was established by summing the monthly total of hotspots in each grid cell of the Cerrado, monthly total burned area by summing the area of all pixels considered as burned area in each grid cell of the Cerrado, monthly average precipitation corresponded to the precipitation estimated by the 3B43 product, and monthly average VCI consisted on calculating the monthly average VCI in each grid cell of the regular grid over the Cerrado. The significance of the spatial correlations described above was also tested using the t-Student test with significance level of 5 %.

## 3 Results and Discussion

Considering the entire Brazilian territory, MODIS fire products detected 5,235,881 hotspots and estimated 1,964,544 $km^2$ burned during the 2002-2015 period. Within this total, 1,904,182 hotspots (approximately 36 %) were detected and 1,358,775 $km^2$ (approximately 69 %) were burned in the area corresponding to the Cerrado, making the biome, respectively, the second and the first Brazilian biome for the occurrence of hotspots and burned area during the period analysed. Despite having approximately half of the area of the Amazon biome, the Cerrado presented only 10 % less hotspots than the Amazon during the period between 2002 and 2015. However, considering the yearly density of hotspots in the Brazilian biomes, hotspots density in the Cerrado (0.067 hotspots $km^{-2}$ $year^{-1}$) is approximately 60 % higher than in the Amazon (0.041 hotspots $km^{-2}$ $year^{-1}$). Regarding burned area, even considering the difference in the area of the biomes, the Cerrado concentrated 69 % of the Brazilian burned area between 2002 and 2015, while the Amazon was responsible for 220,182 $km^2$ of the burned area during 2002-2015, approximately 11 % of the total. These results agree with those found by Moreira de Araújo et al. (2012), who used MOD14, MYD14 and MCD45A1 data to analyse the spatial patterns of hotspots and burned area in the Brazilian territory during the 2002-2010 period and found the highest concentration of hotspots in the Amazon biome, while the Cerrado was responsible for 73 % of the Brazilian burned area. It should be considered that the performance assessment of the MCD45A1 product in dense vegetation areas, such as the Amazon, is not good, where omission errors are frequent, as shown by Roy et al. (2008), Cardozo et al. (2012) and Libonati et al. (2015b). Regarding the Cerrado, MCD45A1 usually presents omission errors related to the detection of small burned area patches, mostly due to the coarse spatial resolution of the product, as shown by Libonati et al. (2015a), who also validated MCD64A1 and AQM burned area datasets. On the other hand, considering only September, when fires are more frequent in the Cerrado, and, therefore, burned area patches tend to be larger, MCD45A1 validation study conducted by Moreira de Araújo and Ferreira (2015) showed good assessment when compared to burned area maps derived from Landsat images.

Considering only the Cerrado, the inter-annual and intra-annual variability in the occurrence of fires, precipitation and VCI along the considered period is shown in Fig. 2. Hotspots and burned area are concentrated during the dry season (May to September for most of the Cerrado), however, they still have high average in October, beginning of the rainy season for most

of the Cerrado, as will be shown below. It is also possible to note the inverse relationship between fire occurrence and precipitation or vegetation condition, once months with higher monthly values of hotspots or burned area had lower values of precipitation or VCI. Monthly total hotspots ranged from 461 to 98,238 and monthly total burned area ranged from 1.75 km$^2$ to 105,338 km$^2$, while monthly average precipitation and VCI ranged, respectively, from 1.5 mm to 370 mm and from 15.9 %

to 78.3 %. Analysing biomass burning, LULCC and the hydrological cycle in the Northern sub-Saharan Africa, a savanna region suffering an intense LULCC process, Ichoku et al. (2016) also found that the seasonal peak of fires is anti-correlated with annual water-cycle indicators, such as precipitation and vegetation greenness, except in humid West Africa, where this situation occurs only during the dry season and burning virtually stops when monthly average precipitation reaches 120 mm. Annual total hotspots in the biome ranged from 53,798 (2009) to 248,911 (2007) and annual total burned area ranged from

19,023 km$^2$ (2009) to 249,982 km$^2$ (2010), while annual precipitation ranged from 1,209 mm (2007) to 1,706 mm (2009) and annual average VCI ranged from 54% (2007) to 62% (2009), both considered normal or good, as proposed by Coleve (2011), who defined VCI values between 0 % and 20 % as extremely dry, between 20 % and 40 % as dry, between 40 % and 60 % as normal, between 60 % and 80 % as good, and between 80 % and 100 % as excellent. It is possible to note that the occurrence of hotspots and burned area was lower in years of higher precipitation and VCI. The year 2009 presented the highest

precipitation and VCI and the lowest total annual hotspots and burned area, while 2007 presented highest occurrence of hotspots and lowest precipitation and VCI. Highest total annual burned area was found in 2010, approximately 15,000 km$^2$ higher than the annual total burned area in 2007.

Highest monthly total hotspots in the entire Cerrado were found in September for all years analysed, except for 2008, when monthly total hotspots in October was 6 % higher than in September and ranged significantly, from 15,537 (September/2009)

to 98,238 (September/2007) hotspots. On the other hand, lowest monthly total hotspots were found in the middle or end of the rainy season, ranging from 461 (February/2002) to 1,182 (January/2010) hotspots. Regarding burned area, highest monthly total burned area was found in August or September, and ranged from 7,449 km$^2$ (August/2009) to 105,338 km$^2$ (September/2010). Lowest monthly total burned area was concentrated in the middle of the rainy season, ranging from 2 km$^2$ (December/2010) to 23 km$^2$ (March/2015). These results are summarized in Table 1.

Accordingly, monthly average hotspots ranged from 1,022 (February) to 47,670 (September) and monthly average burned area ranged from 14 km$^2$ (December) to 38,913 km$^2$ (September), while monthly average precipitation ranged from 10 mm (August) to 257 mm (January) and monthly average VCI from 24 % (September) to 75 % (March). The increase in the occurrence of fires initiates in May, agreeing with the beginning of the dry season, grows steadily and reaches the maximum in September, end of the dry season for most of the Cerrado biome. In October, beginning of the rainy season for most of the Cerrado, when

precipitation (105 mm) is almost four times higher than the average precipitation during the dry season (27 mm) and average VCI increases in 10 % from September to October, monthly average hotspots and burned area start decreasing, but still have high average (24,489 hotspots and 10,403 km$^2$, respectively). Spontaneous combustion, the possibility of fires ignited criminally, the natural occurrence of fires related to lightning and land management practices are the causes of fires in the beginning of the rainy season. For example, Ramos-Neto and Pivello (2000) found that 91 % of the fire events registered at

the Emas National Park, located in the Cerrado, between June/1995 and May/1999 were caused by lightning during the wet season or in the seasonally transitional months. Furthermore, due to the influence of distinct meteorological phenomena, such as the Intertropical Convergence Zone (ITCZ) and Upper Level Cyclonic Vortex disturbances in the Northern of the biome (North of 6° S), the dry season can be displaced ahead in the year (Kayano and Andreoli, 2009). The Central-southern areas

of the Cerrado are mainly controlled by anticyclones and cold fronts, with the dry season characterized by the incursion and settlement of dry air masses over the region, while the rainy season is characterized by local heat convection and the action of the South Atlantic Convergence Zone (Kayano and Andreoli, 2009). Thus, we have a substantial variation of the dry season peak in the Cerrado, which will be empathized in the spatial analysis presented below, highlighting the need for spatialized information in addition to summed/averaged time series in the Cerrado. Throughout the rainy season, precipitation and VCI

elevate and the average of hotspots and burned area decrease: average of hotspots (5,414) and burned area (1,683 km$^2$) in the rainy season is, respectively, 3.62 and 10.14 times lower than the average of hotspots and burned are in the dry season (19,627 hotspots and 17,072 km$^2$, respectively).

Furthermore, monthly total hotspots and burned area in September/2007 and September/2010 (98,238 and 97,573 hotspots, and 96,152 and 105,338 km$^2$, respectively) represent two remarkable episodes, once the total number of hotspots in these single

months is higher than the total hotspots detected by MODIS active fire products in the entire years of 2006, 2008, 2009, 2011 and 2013 and burned area detected was higher than the burned area in the entire years of 2003, 2004, 2005, 2006, 2008, 2009, 2011, 2013, 2014, and 2015. In these months, the physical conditions for the occurrence of fires were extremely favorable: since the beginning of the dry season in 2007 and 2010 monthly average precipitation was lower than the monthly average precipitation for the Cerrado between 2002 and 2015. Average precipitation in the dry season of 2007 and 2010 was,

respectively, 55 % and 68 % of the average precipitation in the Cerrado for the 2002-2015 period (26.7 mm). The drought during the dry season contributed to make the vegetation vulnerable to fires, when most of the areas of the Cerrado in September/2007 and September/2010 presented low values of VCI, lower than 5 %, especially in the South-western region of the biome. Monthly average VCI in the Cerrado for September/2007 and September/2010 was, respectively 25 % and 33 % lower than the monthly average VCI for September (24 %).

Still analysing Fig. 2, we can see the wide variation of the variables within the months; higher displacements in monthly total hotspots and burned area were found during the dry season, especially in August and September, when outliers were identified, such as September of 2007 and 2010. Regarding precipitation, high variability was found during the entire rainy season, especially December and January, and outliers were found in February, May, June and August. VCI inter-annual boxplot presented highest variation in October and November, and outliers were found in February, August, September and October.

Libonati et al. (2015a) also found seasonal variations in median, lower and upper quartiles and extremes values when analysed monthly values of burned area and precipitation for the entire Cerrado area. Regarding trends in Fig. 2 (blue lines in each intra-annual boxplot, which were all significant according to the t-Student test), a slight decrease in monthly total hotspots and in monthly total burned area was found for the 2002-2015 period (decrease of 8.27 % hotspots and 2.67 % km$^2$ in relation to the monthly average of total hotspots and burned area considering the entire 168 months analysed, respectively), agreeing with

results found by Archibald et al. (2010) for Southern Africa, a region also dominated by savanna formations. The negative trend in fire occurrence was previously reported by Archibald (2016) for Africa, explained by the fragmentation of the landscape caused by LULCC, and by Andela et al. (2017), who found a global decline in burned area caused by human activity, including the Cerrado, which, according to the authors, may be related to the increase in livestock, responsible for suppressing

fire activity by reducing fuel loads, or related to changes in fire management practices. Still, the difficulty of MCD45A1 in detecting small size burned area due to the coarse spatial resolution of the product may account for the negative trend found, once smaller burned area patches seem to be a result from the fragmented landscape related to the advance of the agricultural activities over the natural remnants of the Cerrado. Monthly average precipitation trend was also negative in the period (decrease of 2.93 % in relation to the monthly average precipitation considering the entire 168 months analysed), which also

denotes the importance of human activities in the fire regime of the Cerrado, once fire trends were also negative during the period analysed. This negative trend in monthly precipitation was also found by Marcuzzo et al. (2012), who identified a decrease in precipitation in the Middle-west Cerrado for all months, except to March. According to Coelho et al. (2016), the 2002-2015 period is inserted in a greater period of reduction of precipitation in Central and Southeast Brazil, and has been referenced as a drying period in large-scale studies. Opposed to the other variables, monthly average VCI tends to slightly

increase in the Cerrado (increase of 1.21 % in relation to the monthly average VCI considering the entire 168 months analysed), despite the negative trend in precipitation. When analysing vegetation variability and trends in North-eastern Brazil using BFAST, Schucknecht et al. (2013) found similar results: positive trends in NDVI, which is the base of VCI, and negative trends in precipitation, concluding that the direction of vegetation trends frequently do not match with precipitation trends, also suggesting that analysing only precipitation is not enough for explaining vegetation trends.

Regarding LULC, MCD12Q1 product shows that the Cerrado is mostly composed of savannas, as shown in Table 2. On average, 68.32 % of the Cerrado was composed by savannas during the 2002-2013 period (1,391,371 km$^2$), followed by cropland/natural vegetation mosaic (166,990 km$^2$, 8.2 %), woody savannas (138,313 km$^2$, 6.79 %), croplands (116,856 km$^2$, 5.74 %), grasslands (94,799 km$^2$, 4.66 %), evergreen broadleaf forest (85,668 km$^2$, 4.21 %), and other land-uses (42,451 km$^2$, 2.08 %). The spatial distribution of the most frequent land-use estimated by the MCD12Q1 product during the 2002-2013

period is shown in Fig. 1. Comparing the oldest (2002) and the newest (2013) annual MCD12Q1 LULC maps, we can see that despite the increasing anthropic pressure in the Cerrado savanna areas increased in 3.77 %, followed by croplands (1.65 %), while grasslands and woody savannas areas decreased in 3.22 % and 1.02 %, respectively. These results may be caused by the accuracy of the MCD12Q1 product, which, despite the overall accuracy of 75 %, presents confusion between specific classes; savanna areas are usually confused with woody savannas, grasslands, cropland/natural vegetation mosaic and closed

shrublands areas (Friedl et al., 2010). Moreover, comparing the MCD12Q1 results with two Landsat-based LULC mappings for the Cerrado financed by the Brazilian government for the years 2002 (MMA, 2002) and 2013 (INPE, 2015), we have distinct results: while MCD12Q1 shows that 63.58 % and 67.35 % of the Cerrado in 2002 and 2013, respectively, is composed of savannas, Landsat-based mappings found that natural remnants in the Cerrado (which includes forest and savanna formations) declined from 60.5 % to 54.6 % between 2002 and 2013, therefore, in a period of 11 years natural remnants were

reduced in 120,150 km$^2$. Still comparing the LULC maps for the Cerrado provided by MMA (2002) and INPE (2015), agricultural areas increased 1.2 % (24,434 km$^2$) and pasture areas increased in 3 % (61,093 km$^2$) during the 2002-2013 period. According to Pivello (2011), the LULCC process in the Cerrado usually begins with the use of fire in the natural remnants for removing the vegetation, then the cleared area is shifted to planted pasture, subsistence agriculture or industrial farming. While

in subsistence agriculture areas fire is traditionally used for pest control, crop rotation and pasture management, in industrial farming areas, besides the removal of the natural vegetation, fire is also a tool for burning crop residues, and in extensive beef-cattle production areas annual or biannual fires are used for stimulating regrowth of grass in the dry season when forage stocks are low.  Still according to Pivello (2011), in Cerrado conservation areas fires usually are prohibited and no fire management techniques are applied, such as prescribed burnings, which results in enhanced fuel loads and cause, when natural, accidental

or criminal fires happen, more intense fires and larger burned area.

Approximately 72 % of the hotspots detected by the MODIS active fire products (1,369,913) occurred in savannas areas during the 2002-2015 period (Table 2). Followed by savannas, around 9.7 % of the hotspots occurred in woody savannas (185,099), 4.8 % in grasslands areas (91,535), 4.7 % in cropland/natural vegetation mosaic areas (89,978), 3.8 % in evergreen broadleaf areas (72,510), 3.2 % in croplands (61,223), and 1.8 % in areas of other land-uses (33,924).  Nascimento et al. (2011), when

analysing the occurrence of hotspots in the LULC of the Cerrado for the period between May/2008 and May/2009, found 75.6 %, 13.2 %, 11 % and 0.2 % hotspots in the natural remnants, pasture areas, agriculture areas and other land-uses, respectively. The difference between results found and those found by Nascimento et al. (2011) may be caused by the distinct LULC maps used: while we used annual LULC maps from MCD12Q1 product, Nascimento et al. (2011) used the LULC map provided by MMA (2002). Still, Ichoku et al. (2016), who also used MODIS active fires and MCD12Q1 data, found more than 75 % of the

satellite fire detections in the Northern sub-Saharan Region occurring in savanna and woody savannas areas during the 2001-2014 period, which is also suffering the LULCC process from natural areas to croplands.

Regarding the spatial distribution of fires in the biome, Fig. 3(a) shows the total of hotspots detected by MODIS active fire products in the Cerrado during the 2002-2015 period considering a regular grid of 0.25º. The LULCC process in the Cerrado began in the 1970s in the Southern region of the biome, and advanced over the years to the Northern region (Fearnside, 2000).

Therefore, in the Southern of the biome land-use is well settled, once human occupation in these areas is older, and LULCC from natural remnants to other land-uses is not frequent because there are few natural remnants of the Cerrado, causing lower concentration of hotspots. According to INPE (2015), in 2013 São Paulo (SP), Paraná (PR) and Mato Grosso do Sul (MS) states, located in the Southern of the biome and indicated in Fig 3(a), presented only 17 %, 37 %, and 31 % of natural cover in the areas of Cerrado, respectively.  Still, we can see a considerable number of hotspots in the Northern of São Paulo state. In

this region, fires are related to sugarcane pre-harvest burning: Brazil is the world leader in cropping sugarcane and most of the Brazilian sugarcane cultivation areas are located in the Cerrado belonging areas of São Paulo state, responsible for approximately 50 % of the national production (Rudorff et al., 2010).

Over the years, the agricultural frontier advanced from the Center and North of the Mato Grosso state (MT in Fig 3(a)) to the Central North and Northeast of the biome. Grecchi et al. (2014) analysed, between 1985 and 2005, the decrease of natural

remnants in the Cerrado areas of the Mato Grosso state, traditional in the cultivation of soybeans, and concluded that approximately 42 % of the natural remnants of the Cerrado in the state were shifted to agricultural areas during the period of 20 years analysed. Furthermore, the Northern is the current agricultural expansion area in the Cerrado, especially in the Eastern region of Maranhão, Piauí and Tocantins, Western of Bahia (MA, PI, TO and BA in Fig. 3(a), respectively), and in the region known as MATOPIBA (boundary of Maranhão, Tocantins, Piauí and Bahia states). The Maranhão state presented the highest concentration of hotspots, concentrating 4,428 hotspots in only one grid cell (approximately 316 hotspots year[-1]). According to Spera et al. (2016), the MATOPIBA region can be considered an agricultural frontier since the early 2000s, and, opposed to other areas of the Cerrado, does not have a previous land-use related to cattle ranching, therefore, agriculture is advancing over the natural remnants with the use of fire for converting the land-use rather than advancing over previously cleared pasture areas. Moreover, according to INPE (2015), Bahia, Tocantins, Maranhão and Piauí states still had in 2013, respectively, 67 %, 72 %, 72 %, and 83 % of natural cover in the areas of Cerrado, making them potential areas for agricultural expansion. Analysing the spatial pattern of burned area in the Cerrado using AQM burned area product, Libonati et al. (2015b) and Libonati et al. (2016) also found highest concentration in the Northern of the biome. From the spatial distribution of FRP present in Fig. 3(b), we can see that more intense fires are not necessarily located where hotspots are more concentrated. In fact, the intensity of fires is more dependable on climate and fuels conditions than the absolute number of fire detections (Govender et al., 2006; Rissi et al., 2017), and, therefore, fires occurring in the end of the dry season are potentially more intense (Archibald, 2016). Areas where average FRP was higher (Bahia and Piauí states), presenting an average FRP of up to 300 MW, are predominantly constituted of pasture areas or natural remnants composed of grasslands or savannas, and fires from fuels mostly composed of grassy formations tend to be more intense. For example, while a fire in North-American native forest species with a biomass burning rate of 10 kg s[-1] would have a corresponding FRP of 22 MW (Freeborn et al., 2008), a fire in African savanna formations with the same biomass burning rate would have a corresponding FRP of 27 MW (Wooster et al., 2005).

From spatial patterns of the month with highest incidence of hotspots and burned area, and minimum precipitation and VCI in the Cerrado (Fig. 4), we see that analysing only averaged/summed values for the entire biome can hide some patterns. Despite having the highest average of hotspots in September (59.8 %), as discussed above, 19.4 % of the grid cells concentrated the highest incident of hotspots in August, 12.4 % in October, 1.5 % in June, 2.2 % in July, 2.1 % in November and 2.6 % in other months. Central Cerrado showed highest incidence in September, while cells with highest incidence in October concentrated in the Central-eastern of the biome. August peaks were found in the South of the Cerrado, agreeing with the peak of sugarcane harvest, and in the North of the biome. There are areas with different patterns, such as in the Mato Grosso state (MT in Fig 3(a)), which are areas where fire occurrence is low and are not correlated with precipitation or VCI, as will be shown below. For burned area, 52.4 % of the grid cells concentrated highest incidence of burned area in September, followed by 25.5 % in August, 11.8 % in October, 4.3 % in July, 2% in June, 1.4 % in November and 2.6 % in other months. There is an increase in peaks occurring in August and a decrease in September when compared to hotspots, especially observed in the Central-western

of the Cerrado. From Fig. 2, there is a notable value of average monthly total burned area in August, close to the monthly average in September.

In Fig .2, we see that monthly average precipitation is lower in August, however, Fig. 4(c) shows that minimum precipitation occurred during August for only 7 % of the Cerrado. July and June concentrated, respectively, 51.1 % and 40 % of the minimum precipitation in the Cerrado. There is a clear concentration of minimum precipitation during June in the Central-eastern, while minimum precipitation in July is distributed over the Southern, Central-western and Northern of the biome. October and September accounted for, respectively, 1.6 % and 0.3 % of the minimum precipitation in the Cerrado. Regarding VCI, most of the Cerrado (87 %) concentrated minimum VCI in September, followed by 6.7 % in August, 4 % in October and 2.3 % in other months. While minimum VCI in September is spatially distributed over almost all the Cerrado, areas presenting minimum VCI in October or other months concentrated in the extreme Northern of the biome, where the seasonal regime is distinct from the remaining Cerrado due to the ITCZ influence, and areas where minimum VCI occurred in August or other months were found in the Western, a transitional area from the Cerrado to the Amazon biome.  In here, it should be mentioned that not only droughts are responsible for lowering VCI values; among other reasons we can highlight senesce and fire occurrence, therefore, minimum VCI values in September may also result from fires occurred in August or even the synergy of both drought and fires.

 The lag in months between the minimum of precipitation and maximum of hotspots, minimum of precipitation and maximum of burned area, minimum of VCI and maximum of hotspots and minimum of VCI and maximum of burned area is shown in Fig. 5. Hotspots and burned area peak usually occur two or three months after the minimum of precipitation for most of the Cerrado. Two-months lag between hotspots/burned area and precipitation were more often (34.2 % and 30.1 % of the Cerrado, respectively), and presented values very close to three-months lag (32 % and 30 %, respectively). Three-months lag were especially concentrated in the Central-eastern of the Cerrado, where most of the minimum precipitation occur in June. One-month lag areas (17.4 % and 22.4 %, respectively) occurred in the sugarcane areas of São Paulo state and in the areas where hotspots were more concentrated during the 2002-2015 period (Fig. 3(a)), possibly related to the use of fire in the end of the dry season for enhancing the removal of the natural vegetation in cleared areas or stimulating regrowth of pasture for cattle feeding. No lag between hotspots and precipitation and between burned area and precipitation were found, respectively, in 3.7 % and 5 % of the Cerrado, such as in the Mato Grosso do Sul state (MS in Fig. 3(a)), while areas with minimum of precipitation in the beginning of the dry season and maximum of hotspots/burned area in the end of the dry season (four-months lag) accounted for 8.5 % and 5.9 %, especially located in the extreme North of the biome. Other lags occurred in areas where there is a low occurrence of fires and are not correlated with precipitation or VCI, and were found in 4.2 % and 6.6 % of the Cerrado, respectively. While maximum of hotspots and burned area usually occur two or three months after the minimum of precipitation in the Cerrado, maximum of hotspots and burned area are concentrated in the same month when VCI is minimum for most of the Cerrado (59.7 % and 51.4 % of the area of the biome, respectively). One-month lag were also considerable (18 % and 24.2 %, respectively), mostly occurring in the Southern and Northern of the biome, as well as areas where the maximum of hotspots/burned area was found before the minimum VCI (16.3 % and 14.9 %, respectively). Thus, more than 75 % of the

Cerrado presented lag of up to one month between minimum VCI and maximum of hotspots or burned area, showing that VCI is a good indicator of the vegetation condition and its susceptibility to ignition. Areas where maximum hotspots/burned area occurred before minimum VCI may also result from land management techniques, where anthropic action is the main driver for the occurrence of fires, and areas where fires are not usual, such as in the South of the Mato Grosso do Sul state (MS in Fig. 3(a)).

Accordingly, spatial correlations between monthly total hotspots and monthly average precipitation, between monthly total burned area and monthly average precipitation, between monthly total hotspots and monthly average VCI and between monthly total burned area and monthly average VCI are shown in Fig. 6. Results found in Fig. 6 show that hotspots and burned area are better correlated with VCI than precipitation in the Cerrado. The relatively lower correlation obtained for both hotpots and burned area with precipitation values (Figs. 6 (a) and (b)) when compared to VCI values (Figs. 6 (c) and (d)) can be explained by the results shown in Fig. 5, which presented the higher monthly lag between hotspots and burned area with precipitation than the monthly lag between hotspots and burned area with VCI. Correlations between monthly total hotspots and monthly average precipitation, between monthly total burned area and monthly average precipitation, between monthly total hotspots and monthly average VCI and between monthly total burned area and monthly average VCI were, respectively, significant in 83 %, 75 %, 94 % and 94 % of the Cerrado. Spatially, higher correlations for all the four pairs of variables analysed were found in the areas that concentrated most of the hotspots (Fig. 3(a)), located in the Central-north and Northeast of the biome. It is also worth mentioning the good correlations for the four pairs of variables analysed in the Southern of the biome, in the areas related to sugarcane pre-harvest burning. According to Aguiar et al. (2011), more than 100,000 km$^2$ of sugarcane cultivation areas used pre-harvest burning between 2006 and 2011 in São Paulo state, being that pre-harvest burning was higher in years of lower precipitation, such as 2010. Considering the monthly values analysed, hotspots and VCI (Fig. 6(c)) were the better correlated variables, reaching values lower than -0.7 in many grid cells, especially in the Northern of the biome, and burned area was also better correlated with VCI than precipitation. Moreover, it should be mentioned the results recently found by Nogueira et al. (2017), which computed several FDIs in the Brazilian biomes and suggest that the relationship between climate-relate drivers and fires, expressed by FDIs and burned area correlation, is biome-specific to explain the seasonal variation of burned area in the Brazilian biomes, being that FDIs computed from empirical water balances considering a lower soil capacity are more correlated with the seasonal pattern of fire occurrence in the Cerrado. Still considering the results in Fig. 6, we can state that precipitation can be used as a previous indicator for the occurrence of fires, while VCI can be used as an instantaneous indicator of fire risk in the Cerrado, depending on the location within the biome for both cases. Moreover, if we correlate monthly total hotspots or burned area with monthly average precipitation from one, two, three or four months before, there are relative persistent correlation values close to -0.4 up to three-months lag, especially in the North, extreme North and South of the biome, as seen in Fig. 7, showing that the relationship between precipitation and fires needs to consider precipitation from previous months.

Furthermore, van der Werf et al. (2008) analysed the climatic control on the variability of fire in the tropical and subtropical areas of the globe and showed the importance of other factors in controlling the occurrence of fires in the savannas: besides

climate, land-use and grazing also influence the amount of fuel available for burning, therefore, the relationship between fires and precipitation or VCI may not be uniform. Additionally, the authors also point out that in the savannas the seasonal ignition of anthropic fires depends on the land management: prescribed fire are set in the end of the dry season aiming to increase the removal of unwanted plant species and favor resprouting or in the beginning of the dry season to reduce soil depletion and the

probability of uncontrolled wildfires, which may contribute to the results found in this study and by them. Accordingly, Price et al. (2012) investigated the use of prescribed fires in savanna landscapes of Western Arnhem Land (Australia) and concluded that imposing prescribed fires in the beginning of the dry season can substantially reduce burned area and fire severity. For the Cerrado, Rissi et al. (2017) compared fire behaviour in early, mid and late dry season and found that fire intensity is mainly influenced by the combination of dead fuel percentage and fuel load.

However, according to van der Werf et al. (2008), climate may impose limitations for the occurrence of fires in these areas, once drier periods improve the removal of vegetation during the clearing process; fires in the savannas predominantly occur over short grassland vegetation, which development is directly related to the previous wet season. Still, Randerson et al. (2005) positively correlated the severity of the fire season in the Cerrado with the terrestrial water storage in the early wet season, which, according to Chen et al. (2013), suggests that the increase of fuel loads in the Cerrado may expand the occurrence of

fires in the upcoming dry season.

## 4 Conclusions

Data derived from orbital remote sensing were used to characterize the fire, precipitation and vegetation condition regimes in the Cerrado, as well as spatial patterns and correlation between the variables. This approach was successful for establishing patterns and can be used for analysing other vegetated areas of the globe affected by fires.

Time series of monthly values for the entire biome showed that fires in the biome are higher in September and October, according to the end of the dry season and beginning of the rainy season, which can variate within the biome, when the deficit in precipitation and extreme vegetation condition reached maximum indices. However, spatial analysis showed patterns that are not seeing from time series analysis of monthly average/summed values. For example, monthly values indicated minimum precipitation in August, however, spatial analysis identified minimum precipitation in August for only 7 % of the Cerrado.

Therefore, the variation of the variables analysed within the biome must be considered to better describe fire patterns and climate-related fire drivers in the Cerrado.

Besides the increasing anthropic pressure and changes in the LULC of the biome, hotspots and burned area are tending to decrease during the period analysed, probably a result from the fragmented landscape and smaller burned area patches, more difficult to be detected by orbital sensors such as MODIS. Nevertheless, it should be considered that the time series analysed

in the study corresponded to 14 years, therefore, longer time series should be investigated in future studies in order to verify trends found.

We found that 72 % of the hotspots detected in the Cerrado occurred in savanna areas, which is the predominant land-use of biome. However, MCD12Q1 annual LULC maps showed major discrepancies when compared to Landsat-based LULC maps, therefore, future efforts should concentrate on validating the MCD12Q1 product in the Cerrado.

Spatial analysis also showed that more intense fire occurrences were not necessarily found where hotspots were more concentrated (Northern of the biome, the current agricultural frontier of the Cerrado), showing the link between FRP and fuel type and condition more than the absolute number of detections.

Results found qualify VCI as an indicator of the susceptibility of vegetation to ignition, while precipitation can be used as a previous indicator for the occurrence of fires. Still, future studies should especially focus on the anthropic action as a driver for the occurrence of fires in the Cerrado, which seems to be able to change fire patterns in the tropical savannas according to the land management technique imposed.

**Acknowledgements**

We thank the National Counsel of Technological and Scientific Development (CNPq, grant number 162898/2015-0), the Minas Gerais State Research Foundation (FAPEMIG, grant number APQ-01698-14), and the Coordination for the Improvement of Higher Education Personnel (CAPES, grant number 6123/2015-05) for their financial support.

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

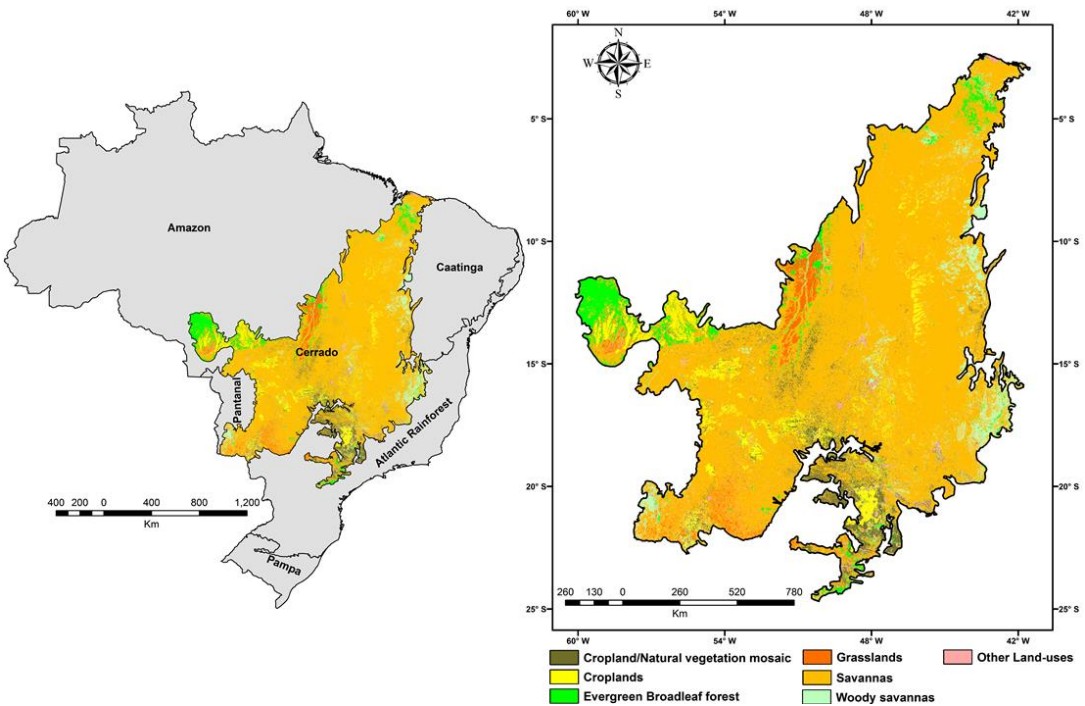

**Figure 1: Location of the Brazilian biomes, highlighting the Cerrado biome. Most frequent LULC in the Cerrado estimated by the MCD12Q1 product during the 2002-2013 period.**

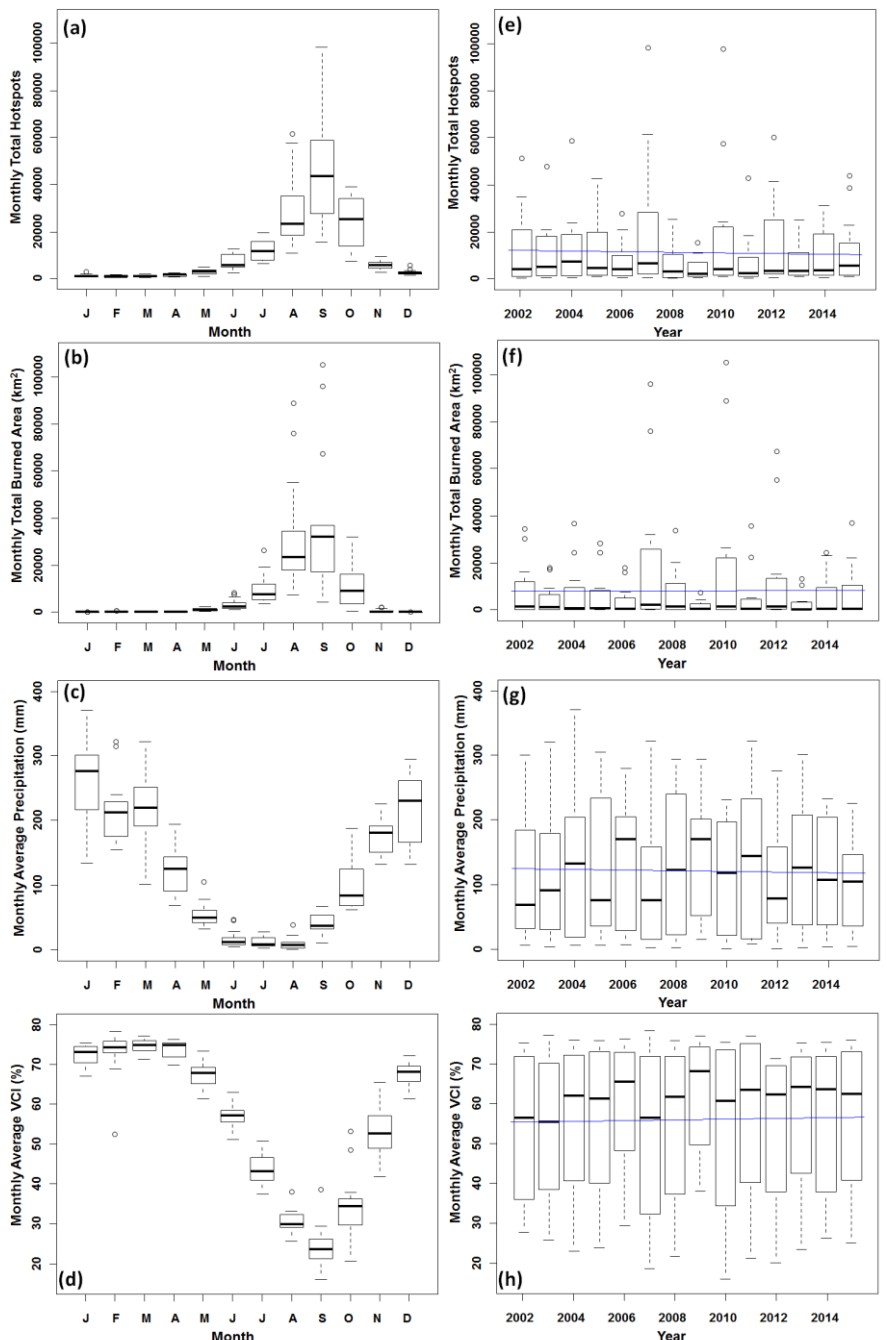

**Figure 2: Inter-annual boxplots for (a) Monthly total hotspots, (b) Monthly total burned area, (c) Monthly average precipitation and (d) Monthly average VCI, and intra-annual boxplots and trends (blue lines) for (e) Monthly total hotspots, (f) Monthly total burned area, (g) Monthly average precipitation and (h) Monthly average VCI in the Cerrado biome during the 2002-2015 period. On each boxplot, the central mark represents the median, the edges of the boxplot are the 25th and 75th percentiles, the upper (lower) whisker delimit the most extreme value contained in the limit defined by the sum (difference) between the 75th (25th) percentile and the difference between the 75th and 25th (25th and 75th) percentiles multiplied by 1.5. Outliers (black circles) represent values outside the limits defined by the whiskers.**

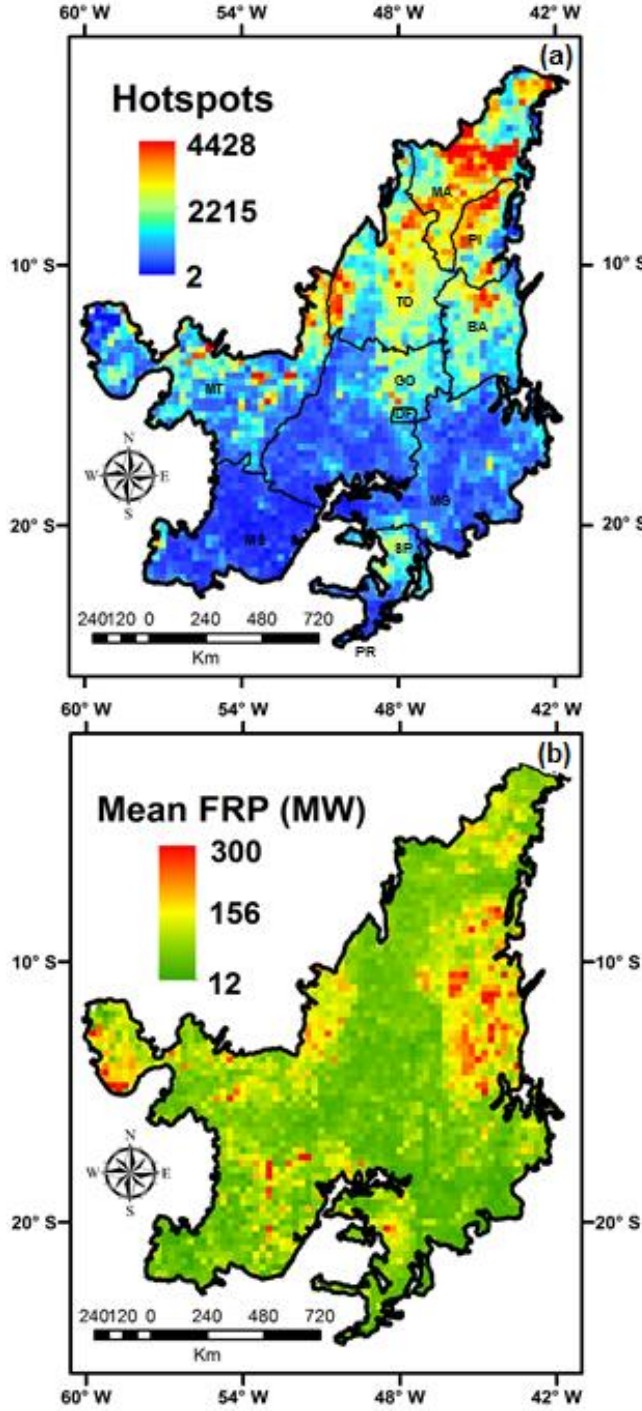

**Figure 3: (a) Total of hotspots and (b) Mean FRP detected by the MODIS active fire products in the Cerrado biome between 2002 and 2015.**

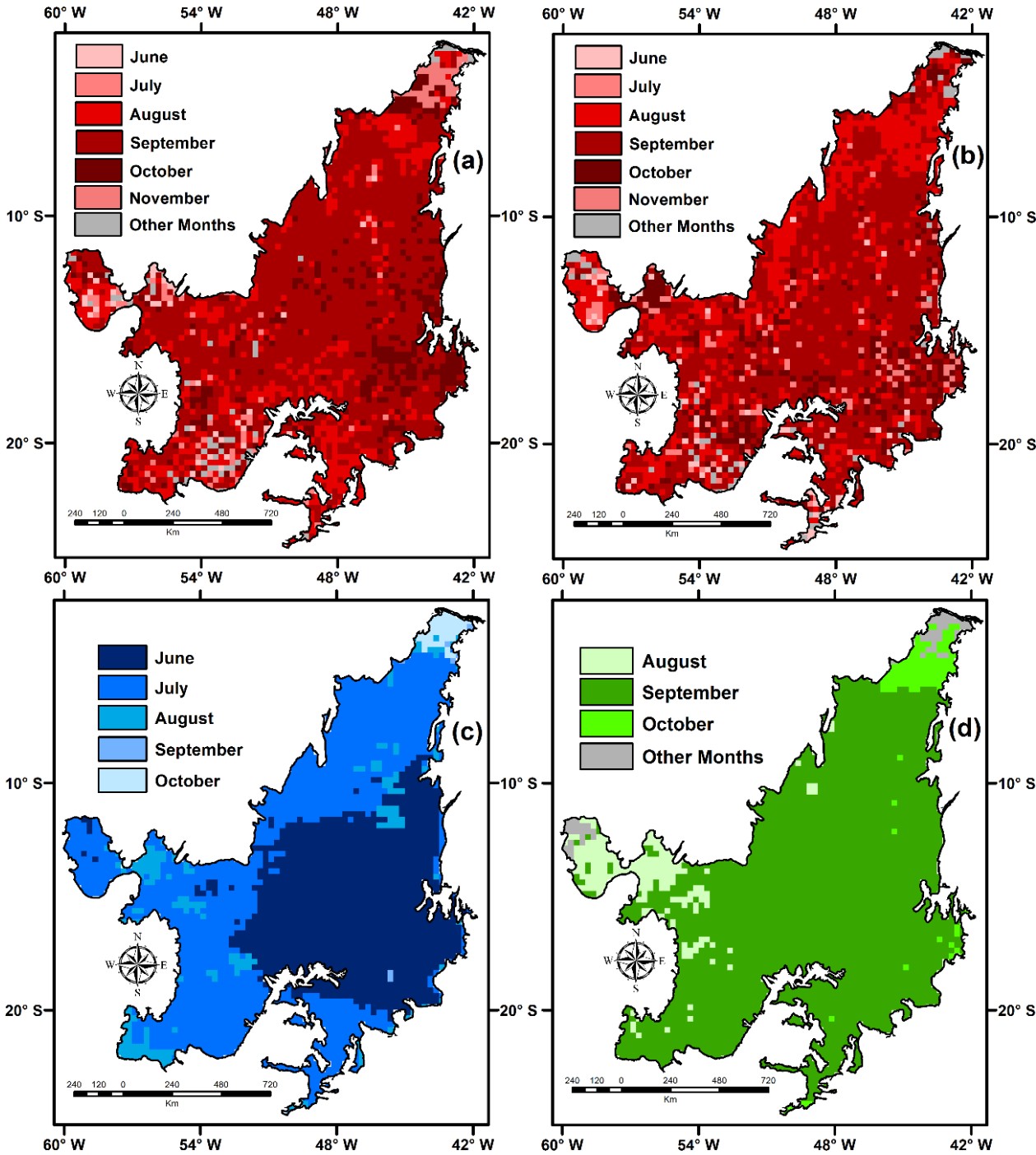

**Figure 4: Estimate of the month when (a) maximum of hotspots (b) maximum of burned area, (c) minimum of precipitation and (d) minimum of VCI was found in the Cerrado for the 2002-2015 time series.**

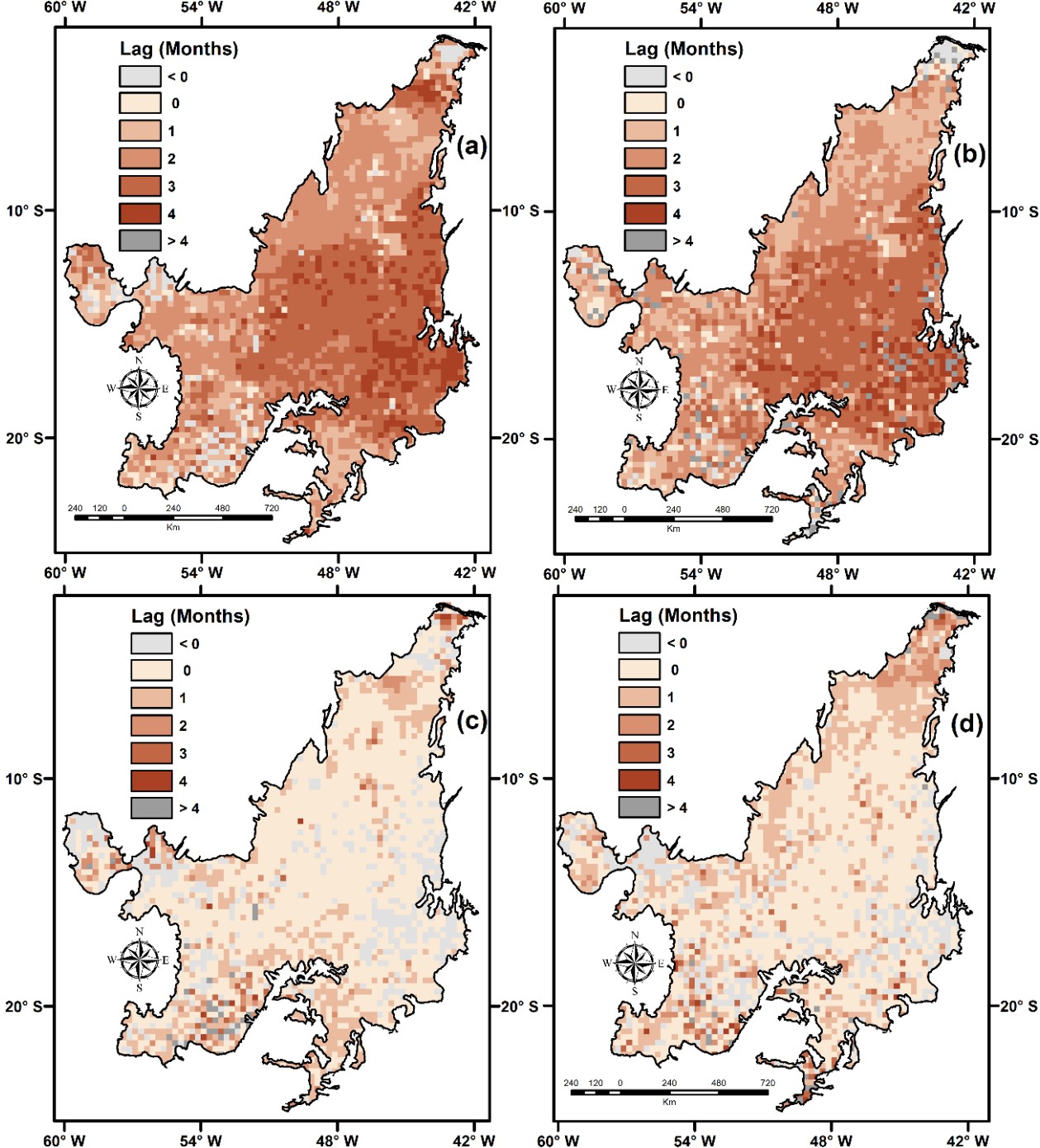

**Figure 5: Lag in months between (a) minimum of precipitation and maximum of hotspots, (b) minimum of precipitation and maximum of burned area, (c) minimum of VCI and maximum of hotspots and (d) minimum of VCI and maximum of burned area in the Cerrado for the 2002-2015 time series.**

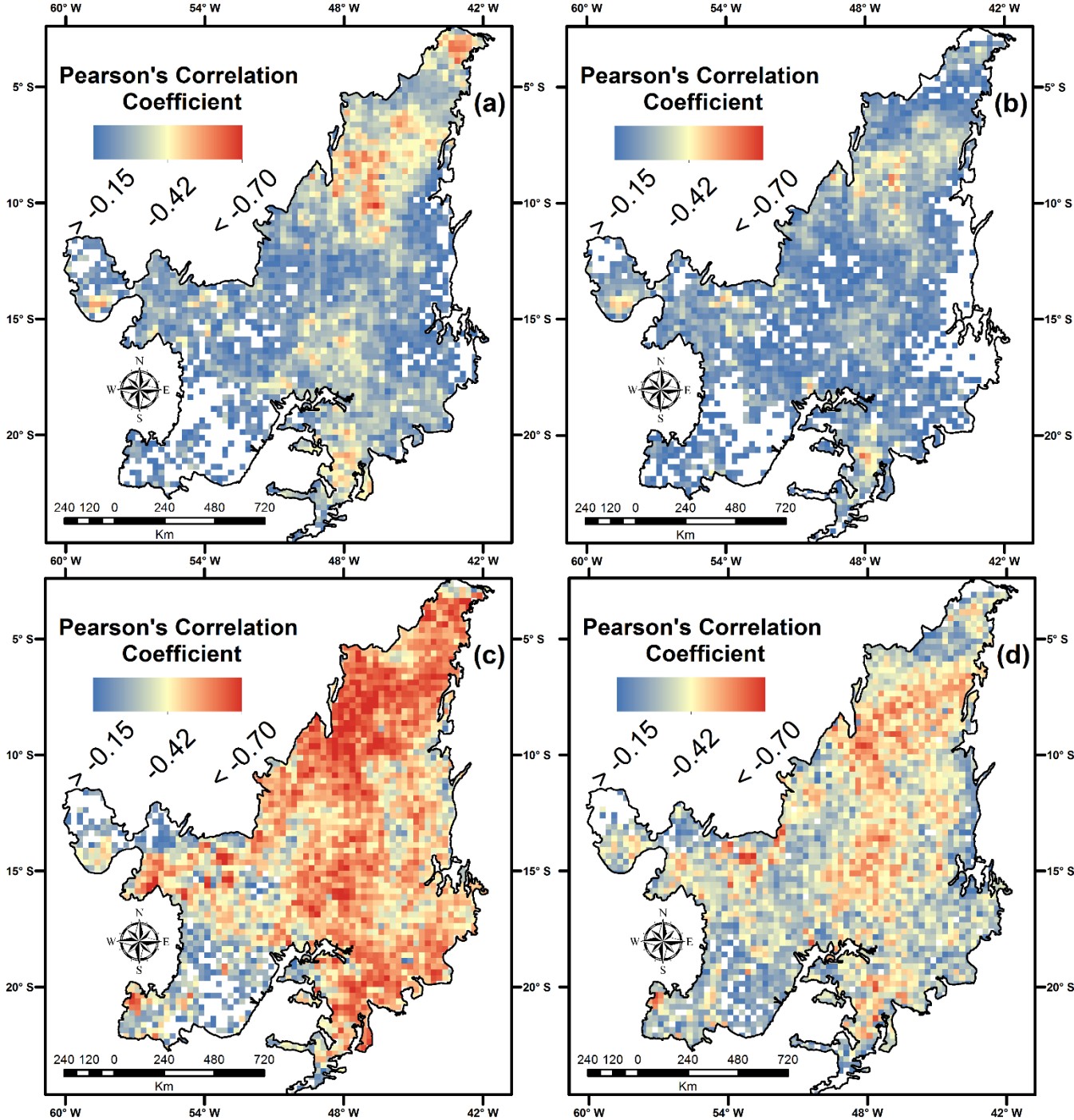

**Figure 6: Spatial correlation between (a) monthly total hotspots and monthly average precipitation, (b) monthly total burned area and monthly average precipitation, (c) monthly total hotspots and monthly average VCI and (d) monthly total burned area and monthly average VCI in the Cerrado biome during the 2002-2015 period. Only statistically significant pixels are shown in the figure.**

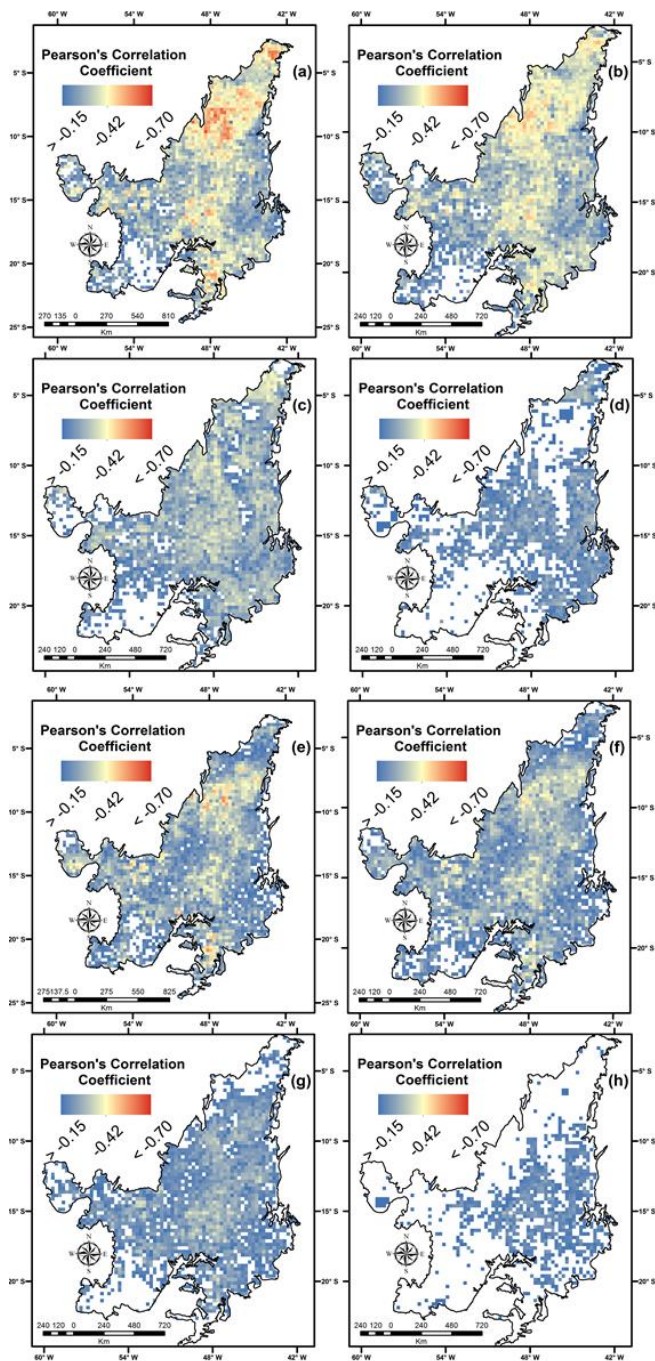

**Figure 7: Spatial correlation between (a) monthly total hotspots and one-month lag monthly average precipitation, (b) monthly total hotspots and two-months lag monthly average precipitation, (c) monthly total hotspots and three-months lag monthly average precipitation, (d) monthly total hotspots and four-months lag monthly average precipitation, (e) monthly total burned area and one-month lag monthly average precipitation, (f) monthly total burned area and two-months lag monthly average precipitation, (g) monthly total burned area and three-months lag monthly average precipitation and (h) monthly total burned area and four-months lag monthly average precipitation in the Cerrado biome. Only statistically significant pixels are shown in the figure.**

**Table 1. Estimate of the month presenting highest monthly total hotspots, lowest monthly total hotspots, highest monthly total burned area and lowest monthly total hotspots and range of the values found in the Cerrado biome for the 2002-2015 time series.**

| Year | Highest Monthly Total Hotspots | Lowest Monthly Total Hotspots | Highest Monthly Total Burned Area | Lowest Monthly Total Hotspots |
|------|------|------|------|------|
| 2002 | September | February | August | December |
| 2003 | September | March | August | December |
| 2004 | September | February | September | January |
| 2005 | September | March | September | December |
| 2006 | September | February | August | December |
| 2007 | September | February | September | December |
| 2008 | August | March | September | January |
| 2009 | September | April | August | December |
| 2010 | September | January | September | December |
| 2011 | September | March | September | March |
| 2012 | September | January | September | December |
| 2013 | September | January | September | December |
| 2014 | September | February | August | December |
| 2015 | September | February | September | March |
| Range | 15,537 - 98,238 | 461 - 1,182 | 7,449 - 150,338 (km$^2$) | 2 - 23 (km$^2$) |

**Table 2. Area range (km$^2$) and average area (km$^2$, %) of the LULC classes in the Cerrado biome estimated by the MCD12Q1 product during the 2002-2013 period, and total and percentage of hotspots detected by the MODIS active fires products in the LULC classes of the Cerrado during the 2002-2015 period.**

| Land-use | Area Range (km$^2$) | Average Area (km$^2$) | Average Area (%) | Total of hotspots | Percentage of Hotspots (%) |
|------|------|------|------|------|------|
| Savannas | 1,294,774 - 1,479,887 | 1,391,371 | 68.32 | 1,369,913 | 71.94 |
| Woody savannas | 106,303 - 169,636 | 138,313 | 6.79 | 185,099 | 9.72 |
| Grasslands | 68,017 - 145,810 | 94,799 | 4.66 | 91,535 | 4.81 |
| Croplands | 102,026 - 146,421 | 116,856 | 5.74 | 61,223 | 3.21 |
| Cropland/Natural vegetation mosaic | 129,111 - 207,514 | 166,990 | 8.20 | 89,978 | 4.73 |
| Evergreen Broadleaf forest | 79,014 - 95,509 | 85,668 | 4.21 | 72,510 | 3.81 |
| Other Land-uses | 38,082 - 49,893 | 42,451 | 2.08 | 33,924 | 1.79 |