# Peer review of "Satellite observations for describing fire patterns and climate-related fire drivers in the Brazilian savannas"

_Natural Hazards and Earth System Sciences, 2017_

## Referee Comment (RC1) · Anonymous Referee #1 · 21 Mar 2017

Title: Analysis of fire dynamics in the Brazilian Savannas This paper analyses the dynamics of fire occurrence in the Brazilian savannas with the aim to understand the occurrence and the dynamics of fires in the Cerrado using precipitation and vegetation condition as explanatory variables. The Cerrado is a key biome for the country and has been heavily transformed over the years, however in the current state, is hard to extract the added value of this paper to for instance Moreira de Araujo et al paper heavily used in the manuscript.

Abstract. From my point of view it fails to state why the subject is important and what is the problem and how it has exactly or partially address. Fires occur and in the lines 10-12 natural and anthropic causes are mentioned, yet the abstract is not clear about these two and mentions the use of data. Results do not mention LULC and no conclusion is presented.

[Figure]

Introduction. In general terms the introduction can be improved. In the same way that the abstract is written, the introduction clearly lacks a presentation of the nature of the problem and both the scope and the state of art regarding seasonality, climatic changes, land use, ignition sources, etc... In this sense paragraph 1 of the introduction needs to develop what is known of both the relationship between natural causes and fires and also human activities. When and where are one more important than others?. What is know about these in particular in Brazil.? As an example also, page 2 L7 the phrase that starts with Moreover,.. instead of only mention role of climatic variables..., the authors should develop what is known, what kind of climatic relationships have been found, regarding total rainfall, seasonality, drought period rainfall, etc? What about interannual and intraannual variability, much is already known about it. It is important to establish what is known to have a better clarity of why this work is important and adds to the current knowledge. Further, not much is included on the relationship between the vegetation conditions previously or during burning season, needs to be developed. In a similar way, page 2 L13-23 lists studies that have used orbital sensors, but why is it important? What those studies have shown? What are the limitations ? in a way that it could eventually lead to state clearly why there is a need for more consistent information (P2L25) which is not clear. Too much of the methods at the end of the introduction, there is no need to state the data used here, leave this for the methods section. No clear objectives or research questions are presented. Finally, so far is difficult to get the value added of this work to others that have already published dynamics of fire occurrence in the region, like some of the cited references for example (Moreira et al 2012, 2015) and others like see (Pivello, 2011)(Chen et al., 2013)

Methods. Not needed so much information on Modis sensors, can be reduced P3L19 onwards P4 L5 is repeated information In general terms dataset used have different spatial resolution, how the authors have used them? Please clarify in the text. I have a strong concern about the methods used to analyze the relationships and I am afraid at this stage Pearson correlation might not be enough and a time series analysis is

needed to capture the complex climate-vegetation conditions-fire occurrence relation-ships. I would suggest the authors to review some literature in relation to this see for instance (Armenteras-Pascual et al., 2011)(Aragão et al., 2008) etc I found the format changes of the Modis products not necessary and basic information that is totally un-needed as it stands (P5 L29-), same with other sections (P6 L15 tiles) etc. P6 L 5-7 The use of a 4 km grid is not justified. P6 L 12 What is most confident? I found extremely confusing the paragraph P7 L3-9, this is the most important part of the methods, to clarify the type of analysis for each research question.

Results and Discussion Since there is no clear research questions, the results are hard to follow but basically the Moreira de Araujo et al 2015 paper shows the same pattern and is heavily used in the discussion, so again the question is what is the value added of this work? I found the fire density reported really high (can be interpreted as every year all km are burnt..), and is confusing if it is over the 2002-2015, I think a yearly density should be calculated or a map of fire frecuency for the period. The LULC and fire relationship is weakly treated. P7 L 30 first time dry season is mentioned, needs to be explained before in the intro or study are a description P8 Lines3-8, consider abbreviating years, summarizing in a table, using ranges when applicable. P8 L 14-15 any other possible explanation? Human ignition? Management practices? P8 L15-17. Is difficult to get where this point originates or relates to the current study. P8L27 consider introducing these general annual trends before monthly patterns P10L2 land us is well settled, not clear the meaning P11L16-21 what about in tropical? And if soil moisture is so important, was not mentioned in the introduction. Figures. Too many, select the most relevant ones, eg. I would remove Fig Conclusions This section needs to be heavily rewritten, first two paragraphs are not conclusions but mostly repetition of results. P12 L 12 How have you established the conclusion of the cerrado as adapted and dependent of fires? This conclusion is not clear, neither in L19-21

Aragão, L. E. O. C., Malhi, Y., Barbier, N., Lima, A., Shimabukuro, Y., Anderson, L., Saatchi, S., Aragao, L. E. O. ., Malhi, Y., Barbier, N., Lima, A., Shimabukuro, Y., Anderson, L. and Saatchi, S.: Interactions between rainfall, deforestation and fires during recent years in the Brazilian Amazonia., Philos. Trans. R. Soc. Lond. B. Biol. Sci., 363(1498), 1779–1785, doi:10.1098/rstb.2007.0026, 2008. Armenteras-Pascual, D., Retana-Alumbreros, J., Molowny-Horas, R., Roman-Cuesta, R. M., Gonzalez-Alonso, F. and Morales-Rivas, M.: Characterising fire spatial pattern interactions with climate and vegetation in Colombia, Agric. For. Meteorol., 151(3), 279–289, doi:10.1016/j.agrformet.2010.11.002, 2011. Chen, Y., Morton, D. C., Jin, Y., Collatz, G. J., Kasibhatla, P. S., van der Werf, G. R., DeFries, R. S. and Randerson, J. T.: Long-term trends and interannual variability of forest, savanna and agricultural fires in South America, Carbon Manag., 4(6), 617–638, doi:10.4155/cmt.13.61, 2013. Pivello, V. R.: The Use of Fire in the Cerrado and Amazonian Rainforests of Brazil: Past and Present, Fire Ecol., 7(1), 24–39, doi:10.4996/fireecology.0701024, 2011.
* * *

---

## Referee Comment (RC2) · Anonymous Referee #2 · 12 Apr 2017

This work addresses an important subject that is the analysis of fire occurrence in the Brazilian Savannah using active fire and burned area information and the relation with climate variables. Accordingly, the authors try to characterize fire regime using vegetation and precipitation data. While the topic is appropriate for NHESS, I cannot recommend because the paper does not show enough novelty, as it is a research based on previously published results without even mention some of them. The authors must be careful when talking about the lack of such specific studies. In fact, there are some studies that show the spatial and temporal patterns of fire and relation between fires and climatic variables in Cerrado. The authors should see the authors they mention (Moreira et al 2012, 2015; Pivello, 2011,) and also papers listed below: The work of Libonati et al., 2015a shows that the intra-annual variability of burnt area over the Brazilian woody savannah mostly relates to the seasonal regime of precipitation.

These authors also show that there is a marked dry season from May to September, characterized by very low precipitation amounts and that, during the dry period, there is a steady displacement towards higher values of the median, lower and upper quartiles and extremes of the distributions of monthly values of burnt area. It is worth mentioning that the manuscript uses the same precipitation data from Libonati et al., 2015a. The work of Libonati et al., 2015b analyzed the results of three currently burned area products derived from MODIS data, namely AQM (INPE), MCD45A1 (NASA) and MCD64A1 (NASA). The procedure is applied to quantify the overall temporal and spatial distribution patterns of burned areas in Brazil for the period 2005 – 2010 and obtained patterns are compared for each Brazilian biome and related to the respective patterns of fire pixels derived from remote sensing. The Cerrado biome was found the one with the largest BA, followed by Caatinga and Amazônia. Estimates of BA over Brazil from AQM, MCD45 and MCD64 products for the period present a similar inter-annual variability. In addition, the work of Libonati et al., 2016 allows analyzing the overall temporal and spatial distribution patterns of BA for the last decade. The highest monthly mean amount is observed in September, followed by October, and March presents the lowest amount. The most severe year is 2007, followed by 2005 and 2010; 2006 and 2009 are the years with less area burned, followed by 2008. The spatial pattern of BA shows that the north region of Cerrado presents the highest frequency of occurrence. The intra and inter-annual variability of BA over Cerrado are closely related to the variability of precipitation but it is worth emphasizing that, despite the major role played by climate conditions, the human factor has also a prominent role on fire dynamics in this region and cannot be disregarded. Accordingly, all the results shown in the manuscript were previously reported by the above studies. However, all these references are missing and the authors argue the novelty of their work. Moreover, I think that there is a gap in the literature review. For instance, the work of Veraverbeke et al., 2014 is no about social and economic costs of fires, but on remote sensing techniques aiming fire severity assessment. In addition, the authors argue the efficiency of MCD45A1 product for mapping and understanding fire behavior and its impacts in the Cerrado. These results are not in agreement with previous works such as Roy et al (2008) and Libonati et al. (2015a), who have pointed out an under-detection of BA by the MCD45 product in savannas regions of Brazil.

Moreover, some conclusions are not based on the results: 1) 'Nevertheless, the annual variability of hotspots and precipitation and between burned area and precipitation during the 2002-2015 period is evidenced.' 2) Besides the seasonal modulation by precipitation, fire occurrence seems to respond to its interannual variability. Drier (wetter) years are associated with more (less) fires in the studied area. There is no clear evidence in the manuscript, despite visual comparison, about the statistical significance of this relation.

The authors say in the conclusions that 'The methods applied are easily implemented and can be used for analyzing the occurrence and dynamics fires in different areas of the globe'. I don't think this work shows a 'method'. Instead, it is mainly devoted to a spatial and seasonal description of available variables.

1) Libonati, et al., 2015a. An Algorithm for Burned Area Detection in the Brazilian Cerrado Using 4 $\mu$m MODIS Imagery. Remote Sensing. , v.7, p.15782 - 15803. 2) Libonati, et al., 2015b. Spatio-temporal variability of burned area over Brazil for the period 2005-2010 using MODIS data In: XVII Simpósio Brasileiro de Sensoriamento Remoto, 2015b, Joao Pessoa. XVII Simpósio Brasileiro de Sensoriamento Remoto. 3) Libonati, et al., 2016. Spatial and temporal patterns of burned area over Brazilian Cerrado from 2005 to 2015 using remote sensing data In: European Geosciences Union General Assembly 2016, 2016, Vienna. European Geosciences Union General Assembly 2016.

---

## Author Comment (AC1) · 14 Jun 2017

Dear Referee 1,

Thank you for your suggestions. These were very helpful and will significantly improve the quality of the revised manuscript. Once there is no option of sending the updated version of the manuscript during this stage of the reviewing process, we will answer each topic indicating the modifications that we made and the new results found.

1) Title. Analysis of fire dynamics in the Brazilian Savannas This paper analyses the dynamics of fire occurrence in the Brazilian savannas with the aim to understand the occurrence and the dynamics of fires in the Cerrado using precipitation and vegetation condition as explanatory variables. The Cerrado is a key biome for the country and has

been heavily transformed over the years, however in the current state, is hard to extract the added value of this paper to for instance Moreira de Araujo et al paper heavily used in the manuscript.

After your considerations, we performed several new analyses, which will be included in the revised paper. The novelty of the manuscript will be empathized below. Most of Moreira de Araújo et al. (2012) is focused on the entire Cerrado, not pixel-based analysis. The Cerrado is a biome distributed over an area of more than 2 million km2 and the relationship between fires and environmental characteristics varies according to several factors. In this work, we used spatial statistical tools to indicate the most vulnerable areas to the occurrence of fires, showing its variation in the biome. Within this context, we consider that, in addition to use a longer time series, we advanced by analysing the correlation between hotspots and burned area with precipitation and VCI spatially (pixel-based), which is not present in Moreira de Araújo et al. (2012) or the other references cited, showing the areas in the Cerrado where the variables are more correlated. Moreover, considering the comments of the Referees, we will add new analysis in the revised paper, described below:

In the updated version of the manuscript we will present a spatial analysis of the month with highest incidence of hotspots and burned area, minimum amount of precipitation and minimum VCI in the Cerrado (Figure 1 in this response letter), as well as the lag in months between the minimum of precipitation and maximum of hotspots, minimum of precipitation and maximum of burned area, minimum of VCI and maximum of hotspots and minimum of VCI and maximum of burned area (Figure 2 in this response letter). Maximum of hotspots and burned area usually occur two or three months after the minimum of precipitation in the Cerrado, while the maximum of hotspots and burned area are concentrated in the same month when VCI is minimum for most of the Cerrado.

In the updated version of the manuscript we will present the seasonality and trend of hotspots, burned area, precipitation and VCI in the Cerrado for the 2002-2015 time series using Breaks For Additive Seasonal and Trend (BFAST), an additive method that

decomposes a time series into seasonal, trend and noise components (VERBESSELT et al., 2010) (Figure 3 in this response letter). A small decrease in the trend of hotspots was found in 2011 and in 2007 for precipitation, while burned area trend was regular during the period and VCI presented a tendency break between 2007 and 2010, showing that VCI is a good indicator of the occurrence of fires in the Cerrado, once 2007 and 2010 were the two years with highest detection of hotspots and burned area in the biome.

In the updated version of the manuscript we will present the mean Fire Radiative Power (FRP) estimated by the MODIS active fire products in the Cerrado between 2002 and 2015 (Figure 4 (b) in this response letter), which showed that more intense fires are not necessarily located where hotspots are more concentrated.

Specific comments:

2) Abstract. From my point of view it fails to state why the subject is important and what is the problem and how it has exactly or partially address. Fires occur and in the lines 10-12 natural and anthropic causes are mentioned, yet the abstract is not clear about these two and mentions the use of data. Results do not mention LULC and no conclusion is presented.

The abstract will be rewritten in the revised paper considering the suggestions of the Referee. Mostly, we will emphasize the importance of fires in the Cerrado, clarify that considering the heterogeneity of the biome a pixel based approach is necessary to understand the complex climate-vegetation conditions-fire occurrence relationships, as well as mention the results regarding LULC, the new results found from the analysis described above and the conclusions obtained in the study.

3) Introduction. In general terms the introduction can be improved. In the same way that the abstract is written, the introduction clearly lacks a presentation of the nature of the problem and both the scope and the state of art regarding seasonality, climatic changes, land use, ignition sources, etc. . . In this sense paragraph 1 of the introduction needs to develop what is known of both the relationship between natural causes and fires and also human activities. When and where are one more important than others?. What is know about these in particular in Brazil.? As an example also, page 2 L7 the phrase that starts with Moreover. . ., instead of only mention role of climatic variables. . ., the authors should develop what is known, what kind of climatic relationships have been found, regarding total rainfall, seasonality, drought period rainfall, etc? What about interannual and intraannual variability, much is already known about it. It is important to establish what is known to have a better clarity of why this work is important and adds to the current knowledge. Further, not much is included on the relationship between the vegetation conditions previously or during burning season, needs to be developed. In a similar way, page 2 L13-23 lists studies that have used orbital sensors, but why is it important? What those studies have shown? What are the limitations ? in a way that it could eventually lead to state clearly why there is a need for more consistent information (P2L25) which is not clear. Too much of the methods at the end of the introduction, there is no need to state the data used here, leave this for the methods section. No clear objectives or research questions are presented. Finally, so far is difficult to get the value added of this work to others that have already published dynamics of fire occurrence in the region, like some of the cited references for example (Moreira et al 2012, 2015) and others like see (Pivello, 2011) (Chen et al., 2013).

The Introduction section will be revised, improved and new references will be cited according to the comments of the Referee, such as Benali et al. (2017), who analysed the extent of the fire regime globally and identified a bimodal seasonality pattern which indicates an anthropogenic fingerprint; Jolly et al. (2015), who studied climate-induced variations in global wildfire danger and found an increasing frequency of longer fire weather seasons in recent years; Leblon et al. (2012), who studied the use of remote sensing in wildfire management and presented an overview of the role of vegetation and weather conditions over the ignition and spread of wildfires; Chéret and Denux (2013), who used NDVI derived from MODIS to estimate the susceptibility of Mediterranean forest to fires; Chen et al. (2011), who studied long-term trends and the inter-annual variability of fires in South America and found large year-to-year changes associated with extreme climate conditions; Pivello (2011), who presented an overview of the fire history in the Amazon and in the Cerrado and described how fire regime changed in the biomes; and Rissi et al. (2017), who compared fire behavior in early, mid and late dry season of the Cerrado and found that fire intensity is mainly influenced by the combination of dead fuel percentage and fuel load. The citation of the studies on P2 L13-23 intended to show that the use of orbital sensors is a widespread approach to understand the role of fire in ecosystems and climate, especially in the savannas around the globe, also showing that MODIS was previously used in regional and global studies of the savannas. In fact, results found were compared with some of the studies presented on P2 L13-23, such as those that are specific for the Cerrado (Nascimento et al., 2010; Moreira de Araújo et al., 2012; Moreira de Araújo and Ferreira, 2015). Moreover, we will remove the statement regarding the data used in the introduction of the revised version of the paper, and clarify the research question and the value added of the paper to others considering the results found and the new analysis proposed in the revised paper.

4) Methods. Not needed so much information on Modis sensors, can be reduced P3L19 onwards P4 L5 is repeated information. In general terms dataset used have different spatial resolution, how the authors have used them? Please clarify in the text. I have a strong concern about the methods used to analyze the relationships and I am afraid at this stage Pearson correlation might not be enough and a time series analysis is needed to capture the complex climate-vegetation conditions-fire occurrence relationships. I would suggest the authors to review some literature in relation to this see for instance (Armenteras-Pascual et al., 2011)(Aragão et al., 2008) etc I found the format changes of the Modis products not necessary and basic information that is totally unneeded as it stands (P5 L29-), same with other sections (P6 L15 tiles) etc. P6 L 5-7 The use of a 4 km grid is not justified. P6 L 12 What is most confident? I found extremely confusing the paragraph P7 L3-9, this is the most important part of the

methods, to clarify the type of analysis for each research question.

We will reduce the information related to MODIS sensors in the section. Regarding the spatial resolution of the datasets, time series of monthly and annual averages presented in Figures 3, 4 and 7 of the Discussion paper considered the entire area of the Cerrado, therefore, values corresponded to the sum of the hotspots and burned area and to the average of precipitation and VCI for the whole biome. Regarding the spatial results, all maps of the revised paper will consider the same grid size (0.25° x 0.25°, spatial resolution of TRMM data), enabling that all results are comparable, therefore, Figures 8, 10 and 11 of the Discussion paper will replaced by Figures 4(a), 5 and 6 of this response letter, respectively. All new Figures were generated using the values of hotspots, burned area, precipitation and VCI corresponding, respectively, to the sum of monthly total hotspots, sum of monthly total burned area, the original TRMM monthly precipitation values and the monthly average VCI for each grid cell of the 0.25° x 0.25° grid over the Cerrado. This will be clarified in the revised paper. Furthermore, considering the comments of the Referee and in order to better understand the complex climate-vegetation conditions-fire occurrence relationships, we will add the new analysis proposed above, which are represented in Figures 1, 2 and 3 of this response letter. All the information considered unnecessary by the Referee will be removed or rewritten in the revised paper. In P6 L12, most confident are pixels flagged as 1 in the MCD45A1 product, which are highly reliable observations, that is, they are the most probable pixels of being burned area. The paragraph P7 L3-9 will be rewritten and the method for estimate the spatial correlations will be clarified, mostly considering the information of the second paragraph of this response letter regarding the Methods section.

5) Results and Discussion Since there is no clear research questions, the results are hard to follow but basically the Moreira de Araujo et al 2015 paper shows the same pattern and is heavily used in the discussion, so again the question is what is the value added of this work? I found the fire density reported really high (can be interpreted as every year all km are burnt), and is confusing if it is over the 2002-2015, I think a yearly

density should be calculated or a map of fire frequency for the period. The LULC and fire relationship is weakly treated. P7 L 30 first time dry season is mentioned, needs to be explained before in the intro or study area description P8 Lines3-8, consider abbreviating years, summarizing in a table, using ranges when applicable. P8 L 14-15 any other possible explanation? Human ignition? Management practices? P8 L15-17. Is difficult to get where this point originates or relates to the current study. P8L27 consider introducing these general annual trends before monthly patterns P10L2 land use is well settled, not clear the meaning P11L16-21 what about in tropical? And if soil moisture is so important, was not mentioned in the introduction.

Considering the new analysis proposed and the comments of the Referee, we think that the value added of the work will be clearer in the revised paper. Fire density reported considered the entire 2002-2015 period, and, in the revised paper, yearly density will be calculated. Additionally, the relationship between LULC and fire will be better explored and discussed, and the dry and rainy season explanation will be moved to the Study Area section. Results beginning in P8 L3-8 will be summarized in a table in the revised paper. In P8 L14-15, precipitation and vegetation conditions due to the accumulated months of drought are the better explanation, however, management practices may also influence and will be cited as a possible explanation in the revised paper. As a result of the new analysis (Figures 1 and 2 of this response letter), we can see that spatially there is a variation in the maximum values of hotspots and burned area and minimum precipitation and VCI and in the lag between the variables, which helps to understand why there are still high averages in October. These results will be better explored in the revised version of the paper. Additionally, P8 L15-17 will be rewritten. The suggestion regarding P8 L27 will be considered in the revised paper. In P10 L2 land use well settled means that land use change is not usual in recent days, once human occupation in these areas is older and there are few natural remnants of the Cerrado. The citations in P11 L16-21 intended to present other climate controllers that have influence over fires in other vegetated areas of the globe and may also influence the occurrence of fires in the Cerrado, but were not analysed yet. Considering the

comment of the Referee and once these variables were not tested for the Cerrado we will remove the P11 L16-21 from the revised paper.

6) Conclusions. This section needs to be heavily rewritten, first two paragraphs are not conclusions but mostly repetition of results. P12 L 12 How have you established the conclusion of the Cerrado as adapted and dependent of fires? This conclusion is not clear, neither in L19-21

The Conclusions section will be entirely rewritten considering the comments of the Referee and the new results found, especially the first two paragraphs. The new Conclusions section will include the following topics regarding the new analysis proposed:

Analysing only average values are not the best approach to characterize the occurrence of fires in the Cerrado;

Spatial analysis and its relationship with the variation of hotspots, burned area, precipitation and VCI in the Cerrado;

Usually, there is a lag of 2 or 3 months between the minimum values of precipitation and hotspots/burned area in the Cerrado and no lag between VCI and hotspots/burned area in the biome;

A statement regarding VCI as a good indicator of the occurrence of fires in the Cerrado;

More intense fires are not located in the areas where hotspots are more concentrated in the Cerrado.

7) Figures. Too many, select the most relevant ones, eg. I would remove Fig

After several changes and new analysis, the number of figures will be reduced, such as Figure 5 of the Discussion paper.

New references:

Benali, A., Mota, B., Carvalhais, N., Oom, D., Miller, L. M., Campagnolo, M. L.,

Pereira, J. M. C.: Bimodal fire regimes unveil a global-scale anthropogenic finger-print, Glob. Ecol. Biogeo., doi: 10.1111/geb.12586, 2017. Chen, Y., Morton, D. C., Jin, Y., Collatz, G. J., Kasibhatla, P. S., van der Werf, G. R., DeFries, R. S., and 5 Randerson, J. T.: Long-term trends and interannual variability of forest, savanna and agricultural fires in South America, Carbon Manag., 4, 6, doi: 10.4155/cmt.13.61, 2014. Chéret, V., Denux, J. P.: Analysis of MODIS NDVI Time Series to Calculate Indicators of Mediterranean Forest Fire Susceptibility, GISci. Rem, Sens., 48, 2, doi: 10.2747/1548-1603.48.2.171, 2013. Joly, W. M., Cochrane, M. A., Freeborn, P. H., Holden, Z. A., Brown, T. J., Williamson, G. J., Bowman, D. M. J. S.: Climate-induced variations in global wildfire danger from 1979 to 2013, Nature Comms, 6, doi: 10.1038/ncomms8537, 2015. Leblon, B., Bourgeau-Chavez, L., San-Miguel-Ayanz, J.: Use of Remote Sensing in Wildfire Management, in Sustainable Development - Authoritative and Leading Edge Content for Environmental Management, 1 st edition, InTech, Press, Rijeka, Croatia, 55-82, 2012. Pivello, V. R.: The use of fire in the Cerrado and Amazonian rainforests of Brazil: past and present, Fire Ecol., 7, 1, doi: 10.4996/fireecology.0701024, 2011. Rissi, M. N., Baeza, M. J., Gorfone-Barbosa, E., Zupo, T., Fidelis, A.: Does season affect fire behaviour in the Cerrado?, Int. J. Wildland Fire, 26, 5, doi: 10.1071/WF14210, 2017. Verbesselt, J., Hyndman, R., Newnham, G., Culvenor, D.: Detecting trend and seasonal changes in satellite image time series, Rem. Sens. Env., 114, 1, doi: j.rse.2009.08.014, 2010.

[Figure]

**Fig. 1.** Estimate of the month when (a) maximum of hotspots (b) maximum of burned area, (c) minimum of precipitation and (d) minimum of VCI was found in the Cerrado for the 2002-2015 time series.

[Figure]

**Fig. 2.** Lag in months between minimum and maximum values of (a) precipitation and hotspots, (b) precipitation and burned area, (c) VCI and hotspots and (d) VCI and burned area in the Cerrado.

[Figure]

**Fig. 3.** Decomposition of the (a) hotspots, (b) burned area, (c) precipitation and (d) VCI time series in the Cerrado (Yt) into seasonality (St), Trend (Tt) and Remainder (et) components.

[Figure]

**Fig. 4.** (a) Total of hotspots and (b) Mean FRP detected by the MODIS active fire products in the Cerrado biome between 2002 and 2015.

[Figure]

**Fig. 5.** Spatial correlation between (a) hotspots and precipitation, (b) burned area and precipitation, (c) hotspots and VCI and (d) burned area VCI in the Cerrado biome.

[Figure]

**Fig. 6.** Spatial t-Student test for the spatial correlation between (a) hotspots and precipitation, (b) burned area and precipitation, (c) hotspots and VCI and (d) burned area VCI in the Cerrado biome.

---

## Author Comment (AC2) · 14 Jun 2017

Dear Referee 2,

Thank you for your suggestions. These were very helpful and will significantly improve the quality of the revised manuscript. Once there is no option of sending the updated version of the manuscript during this stage of the reviewing process, we will answer each topic indicating the modifications that we made and the new results found.

1) The authors must be careful when talking about the lack of such specific studies. In fact, there are some studies that show the spatial and temporal patterns of fire and relation between fires and climatic variables in Cerrado. The authors should see the authors they mention (Moreira et al 2012, 2015; Pivello, 2011,) and also papers listed

below: The work of Libonati et al., 2015a shows that the intra-annual variability of burnt area over the Brazilian woody savannah mostly relates to the seasonal regime of precipitation. These authors also show that there is a marked dry season from May to September, characterized by very low precipitation amounts and that, during the dry period, there is a steady displacement towards higher values of the median, lower and upper quartiles and extremes of the distributions of monthly values of burnt area. The work of Libonati et al., 2015b analyzed the results of three currently burned area products derived from MODIS data, namely AQM (INPE), MCD45A1 (NASA) and MCD64A1 (NASA). The procedure is applied to quantify the overall temporal and spatial distribution patterns of burned areas in Brazil for the period 2005 – 2010 and obtained patterns are compared for each Brazilian biome and related to the respective patterns of fire pixels derived from remote sensing. The Cerrado biome was found the one with the largest BA, followed by Caatinga and Amazônia. Estimates of BA over Brazil from AQM, MCD45 and MCD64 products for the period present a similar inter-annual variability. In addition, the work of Libonati et al., 2016 allows analyzing the overall temporal and spatial distribution patterns of BA for the last decade. The highest monthly mean amount is observed in September, followed by October, and March presents the lowest amount. The most severe year is 2007, followed by 2005 and 2010; 2006 and 2009 are the years with less area burned, followed by 2008. The spatial pattern of BA shows that the north region of Cerrado presents the highest frequency of occurrence. The spatial pattern of BA shows that the north region of Cerrado presents the highest frequency of occurrence. The intra and inter-annual variability of BA over Cerrado are closely related to the variability of precipitation but it is worth emphasizing that, despite the major role played by climate conditions, the human factor has also a prominent role on fire dynamics in this region and cannot be disregarded.

This paper is not focused on Burned Area (BA) or how to estimate it. We used this variable as an additional dataset to spatially analyse fire patterns over the Cerrado. Most of the articles cited by the Referee use averaged/summed values for the entire Cerrado and are focused only in BA and fire count. In the present study, we analysed several

sources at pixel-based statistics, such as spatial correlation. The Cerrado is a biome distributed over an area of more than 2 million km2 and the relationship between fires and environmental characteristics varies according to several factors. Within this context, we used spatial statistical tools to indicate the most vulnerable areas, showing its variation in the biome, which will be emphasized in the revised paper. We will include the references of Libonatti et al. in the revised paper when discussing results found for BA. Regarding the lack of studies, this sentence will be corrected in the revised paper, referring to spatial analysis. We meant that there is a lack of studies analysing the correlation between hotspots/BA and climatic variables spatially, such as the results presented in Figure 10 of the Discussion paper and in the new analysis proposed below. Pivello (2011), Moreira de Araújo et al. (2012), Moreira de Araújo and Ferreira (2015), and Libonati et al. (2015a, 2015b, and 2016) did not provide this information. In fact, all the works cited above, except to Pivello (2011), who presented an overview of the fire history in the Amazon and in the Cerrado and described how fire regime changed in the biomes, are focused on BA, which was only one of the variables analysed and discussed in the paper. Two of the references are focused on describing an algorithm for BA detection and validating the MCD45A1 BA product (Libonati et al. (2015a) and Moreira de Araújo and Ferreira (2015), respectively). Moreira de Araújo et al. (2012) and Libonatti et al. (2015a) analysed the correlation between BA and precipitation in the Cerrado, however, not spatially, therefore, they did not show in which areas of the biome these two variables are more correlated. Libonatti et al (2015b), which was published in the proceedings of the Brazilian Symposium on Remote Sensing, compared 3 different BA datasets (AQM, MCD45A1, and MCD64A1) and also analysed the spatial and temporal variability of BA in the Brazilian territory for the 2005-2010 period. Their work did not analyse the spatial and temporal variability in the area corresponding to the delimitation of the Cerrado, only nationally. We analysed the temporal distribution of hotspots and BA in the Cerrado considering a longer time series (2002-2015), as well as the spatial distribution of hotspots in the Cerrado (Figure 8 of the Discussion paper). Regarding Libonati et al. (2016), we found only a one-page abstract of the ref-

erence published in the proceedings of the EGU General Assembly. The results cited by the Referee were obtained from AQM product, are described in one paragraph and do not show the annual or monthly values of BA or the map showing that the highest concentration of BA is in the North of the Cerrado. Moreover, we have substantially discussed the role of human activities in the occurrence of fires in the savannas (from P11 L23 to the end of the Results and Discussion section).

2) It is worth mentioning that the manuscript uses the same precipitation data from Libonati et al., 2015a.

TRMM is the most used dataset in studies of precipitation conducted using remote sensing, once it provides excellent estimation of spatial and temporal patterns of precipitation considering a period of more than 15 years and is widely validated. The efficiency of TRMM data in the Brazilian territory is shown in works such as Pereira et al. (2013), cited in the Discussion paper, which justifies the choice of the dataset. In fact, Libonati et al. (2015a) was not the first paper that used TRMM dataset for analysing precipitation in the Cerrado. We can find several references that use the same product, for example, this reference from 2011 related to the environmental analysis in South America using TRMM and fire datasets (http://www.mdpi.com/2072-4292/3/10/2110), which is prior to the mentioned study. Furthermore, Moreira de Araújo et al. (2012) also used TRMM data when analysed the distribution patterns of burned area in the Brazilian biomes. Moreover, Moreira de Araújo et al. (2012) and Libonati et al. (2015a) used an average value for the entire Cerrado and did not considered the spatial variability of precipitation, which includes distinct mechanisms. In the Cerrado, we have a substantial variation of the dry season peak, for example, the North region of the biome is mainly controlled by the Intertropical Convergence Zone and Upper Level Cyclonic Vortex disturbances, while the South region of the biome is mainly controlled by anticyclones and cold fronts. Thus, using only average values for the entire biome could not be the best approach. Therefore, we proposed to use a spatial approach with statistical analysis pixel-by-pixel, which will be described below.

3) Accordingly, all the results shown in the manuscript were previously reported by the above studies.

Considering the comments of the Referees and in order to improve the novelty of the work, we will add new analysis in the revised paper, described below:

In the updated version of the manuscript we will present a spatial analysis of the month with highest incidence of hotspots and burned area, minimum amount of precipitation and minimum VCI in the Cerrado (Figure 1 in this response letter), as well as the lag in months between the minimum of precipitation and maximum of hotspots, minimum of precipitation and maximum of burned area, minimum of VCI and maximum of hotspots and minimum of VCI and maximum of burned area (Figure 2 in this response letter). Maximum of hotspots and burned area usually occur two or three months after the minimum of precipitation in the Cerrado, while the maximum of hotspots and burned area are concentrated in the same month when VCI is minimum for most of the Cerrado.

In the updated version of the manuscript we will present the seasonality and trend of hotspots, burned area, precipitation and VCI in the Cerrado for the 2002-2015 time series using Breaks For Additive Seasonal and Trend (BFAST), an additive method that decomposes a time series into seasonal, trend and noise components (VERBESSELT et al., 2010) (Figure 3 in this response letter). A small decrease in the trend of hotspots was found in 2011 and in 2007 for precipitation, while burned area trend was regular during the period and VCI presented a tendency break between 2007 and 2010, showing that VCI is a good indicator of the occurrence of fires in the Cerrado, once 2007 and 2010 were the two years with highest detection of hotspots and burned area in the biome.

In the updated version of the manuscript we will present the mean Fire Radiative Power (FRP) estimated by the MODIS active fire products in the Cerrado between 2002 and 2015 (Figure 4 (b) in this response letter), which showed that more intense fires are not necessarily located where hotspots are more concentrated.

Moreover, all maps of the revised paper will consider the same grid size (0.25° x 0.25°, spatial resolution of TRMM data,), enabling that all results are comparable, therefore, Figures 8, 10 and 11 of the Discussion paper will be replaced by Figures 4(a), 5 and 6 of this response letter, respectively.

4) For instance, the work of Veraverbeke et al. (2014) is no about social and economic costs of fires, but on remote sensing techniques aiming fire severity assessment. In addition, the authors argue the efficiency of MCD45A1 product for mapping and understanding fire behavior and its impacts in the Cerrado. These results are not in agreement with previous works such as Roy et al (2008) and Libonati et al. (2015a), who have pointed out an under-detection of BA by the MCD45 product in savannas regions of Brazil.

The citation of the social and economic costs of fires will be corrected in the revised paper. Regarding the efficiency of the MCD45A1 product for mapping and understanding fire behavior and its impacts in the Cerrado, this sentence will be rewritten, and the work of Libonati et al. (2015a) will be cited in the revised paper. Moreira de Araújo and Ferreira (2015) analysed the performance assessment of the BA product MCD45A1 in the Cerrado by comparing the product with BA maps derived from Landsat images and found good results, however, they analysed only BA detected during September, which is the month with highest concentration of BA in the Cerrado.

5) Some conclusions are not based on the results: 1) 'Nevertheless, the annual variability of hotspots and precipitation and between burned area and precipitation during the 2002-2015 period is evidenced.' 2) Besides the seasonal modulation by precipitation, fire occurrence seems to respond to its interannual variability. Drier (wetter) years are associated with more (less) fires in the studied area. There is no clear evidence in the manuscript, despite visual comparison, about the statistical significance of this relation. The authors say in the conclusions that 'The methods applied are easily implemented and can be used for analyzing the occurrence and dynamics fires in different areas of the globe'. I don't think this work shows a 'method'. Instead, it is

mainly devoted to a spatial and seasonal description of available variables.

The Conclusions section will be entirely rewritten considering the comments of the Referee and the new results found, especially the first two paragraphs. The new Conclusions section will include the following topics regarding the new analysis proposed:

Analysing only average values are not the best approach to characterize the occurrence of fires in the Cerrado;

Spatial analysis and its relationship with the variation of hotspots, burned area, precipitation and VCI in the Cerrado;

Usually, there is a lag of 2 or 3 months between the minimum values of precipitation and hotspots/burned area in the Cerrado and no lag between VCI and hotspots/burned area in the biome;

A statement regarding VCI as a good indicator of the occurrence of fires in the Cerrado;

More intense fires are not located in the areas where hotspots are more concentrated in the Cerrado.

New reference:

Verbesselt, J., Hyndman, R., Newnham, G., Culvenor, D.: Detecting trend and seasonal changes in satellite image time series, Rem. Sens. Env., 114, 1, doi: j.rse.2009.08.014, 2010.

**Fig. 1.** Estimate of the month when (a) maximum of hotspots (b) maximum of burned area, (c) minimum of precipitation and (d) minimum of VCI was found in the Cerrado for the 2002-2015 time series.

**Fig. 2.** Lag in months between minimum and maximum values of (a) precipitation and hotspots, (b) precipitation and burned area, (c) VCI and hotspots and (d) VCI and burned area in the Cerrado.

**Fig. 3.** Decomposition of the (a) hotspots, (b) burned area, (c) precipitation and (d) VCI time series in the Cerrado (Yt) into seasonality (St), Trend (Tt) and Remainder (et) components.

**Fig. 4.** (a) Total of hotspots and (b) Mean FRP detected by the MODIS active fire products in the Cerrado biome between 2002 and 2015.

**Fig. 5.** Spatial correlation between (a) hotspots and precipitation, (b) burned area and precipitation, (c) hotspots and VCI and (d) burned area VCI in the Cerrado biome.

**Fig. 6.** Spatial t-Student test for the spatial correlation between (a) hotspots and precipitation, (b) burned area and precipitation, (c) hotspots and VCI and (d) burned area VCI in the Cerrado biome

---

## Author Response (AR1)

Dear Editor and Referees,

Thank you for your comments regarding the manuscript. These were valuable suggestions for revising and improving the quality of the revised manuscript. After your considerations, we performed several changes and new analysis, and clarified the novelty of the manuscript, as will be empathized below. The Cerrado is a biome distributed over an area of more than 2 million km$^2$ and the relationship between fires and environmental drivers varies within the biome, highlighting the importance of characterizing the spatial patterns of fire occurrence and their correlation with climatic variables and vegetation condition. Within this context, we consider that, in addition to use a longer time series, we advanced by analysing the correlation between hotspots and burned area with precipitation and VCI spatially (pixel-based), which is not present in Moreira de Araújo et al. (2012) or the other references cited, showing the areas within the Cerrado where the variables are more correlated.

New analysis performed on the revised manuscript are listed below:

- Boxplot and trend analysis of the 2002-2015 time series of monthly total hotspots, monthly total burned area, monthly average precipitation and monthly average VCI (Fig. 4 of the new manuscript);
- Spatial analysis of the mean Fire Radiative Power (FRP) estimated by the MODIS active fire products in the Cerrado between 2002 and 2015 (Fig. 5(b) of the new manuscript);
- Spatial analysis of the month with highest incidence of hotspots and burned area, minimum amount of precipitation and minimum VCI in the Cerrado (Fig. 6 of the new manuscript), as well as the lag in months between the minimum of precipitation and maximum of hotspots, minimum of precipitation and maximum of burned area, minimum of VCI and maximum of hotspots and minimum of VCI and maximum of burned area (Fig. 7 of the new manuscript).
- Spatial correlation of the monthly total hotspots and burned area with monthly average precipitation from one, two, three and four months before.

Moreover, we have substantially rewritten the manuscript considering the comments on the review reports. Main changes performed are described below:

- More than twenty new references were cited in the updated version of the manuscript;
- A new title was proposed for the new version of the manuscript: Satellite observations for describing fire patterns and climate-related fire drivers in the Brazilian savannas;
- Abstract section was entirely rewritten;
- Introduction section was improved, clarifying the research question. Eleven new references were cited in the revised Introduction section;
- Conclusion section was substantially improved, considering both review reports and the new analysis performed.

Regarding the specific comments of the Referees, they are properly replied below:

**Abstract. From my point of view it fails to state why the subject is important and what is the problem and how it has exactly or partially address. Fires occur and in the lines 10-12 natural and anthropic causes are mentioned, yet the abstract is not clear about these two and mentions the use of data. Results do not mention LULC and no conclusion is presented.**

The abstract was rewritten in the revised paper considering the suggestions of the Referee. Mostly, we focused on clarifying the research question, removed the statements regarding the data used, mentioned the results regarding LULC, results found from the new analysis performed and presented some of the conclusions obtained from the study.

**Introduction. In general terms the introduction can be improved. In the same way that the abstract is written, the introduction clearly lacks a presentation of the nature of the problem and both the scope and the state of art regarding seasonality, climatic changes, land use, ignition sources, etc... In this sense paragraph 1 of the introduction needs to develop what is known of both the relationship between natural causes and fires and also human activities. When and where are one more important than others?. What is know about these in particular in Brazil.? As an example also, page 2 L7 the phrase that starts with Moreover..., instead of only mention role of climatic variables..., the authors should develop what is known, what kind of climatic relationships have been found, regarding total rainfall, seasonality, drought period rainfall, etc? What about interannual and intraannual variability, much is already known about it. It is important to establish what is known to have a better clarity of why this work is important and adds to the current knowledge. Further, not much is included on the relationship between the vegetation conditions previously or during burning season, needs to be developed. In a similar way, page 2 L13-23 lists studies that have used orbital sensors, but why is it important? What those studies have shown? What are the limitations ? in a way that it could eventually lead to state clearly why there is a need for more consistent information (P2L25) which is not clear. Too much of the methods at the end of the introduction, there is no need to state the data used here, leave this for the methods section. No clear objectives or research questions are presented. Finally, so far is difficult to get the value added of this work to others that have already published dynamics of fire occurrence in the region, like some of the cited references for example (Moreira et al 2012, 2015) and others like see (Pivello, 2011) (Chen et al., 2013).**

The Introduction section was entirely rewritten, considering the very helpful comments of the Referee. In the section of the revised manuscript, eleven new references were cited. For example, papers were cited regarding the frequency of fire weather seasons (Jolly et al., 2015), the bimodal pattern of fire occurrence (Benali et al., 2017), the role of climate and vegetation condition in the occurrence of fires (Archibald et al., 2010; Chéret and Denux, 2011; Leblon et al., 2012; Chen et al., 2013; Bajocco et al., 2015), the role of fire in the LULCC process of the Cerrado (Pivello, 2011), and other works that analysed the temporal and spatial distribution of fires in the Cerrado (Libonati et al., 2015a; Libonati et al., 2015b; Libonati et al., 2016).

The citation of the studies on P2 L13-23 intended to show that the use of orbital sensors is a widespread approach to understand the role of fire in ecosystems and climate, especially in the savannas around the globe, also showing that MODIS was previously used in regional and global studies of the savannas. In fact, results found were compared with some of the studies presented on P2 L13-23 of the Discussion paper, such as those that are specific for the Cerrado (Nascimento et al., 2010; Moreira de Araújo et al., 2012; Moreira de Araújo and Ferreira, 2015). Nevertheless, we reduced the number of citations in this statement of the new Introduction section. Moreover, we removed the statement regarding the method and reduced the information regarding the data used.

Finally, we considered that the changes applied to the Introduction section were able to better clarify the research question, especially the paragraph starting at P2-L32, as well as the objective, which is present in the last paragraph of the section.

**Methods. Not needed so much information on Modis sensors, can be reduced P3L19 onwards P4 L5 is repeated information. In general terms dataset used have different spatial resolution, how the authors have used them? Please clarify in the text. I have a strong concern about the methods used to analyze the relationships and I am afraid at this stage Pearson correlation might not be enough and a time series analysis is needed to capture the complex climate-vegetation conditions-fire occurrence relationships. I would suggest the authors to review some literature in relation to this see for instance (Armenteras-Pascual et al., 2011)(Aragão et al., 2008) etc I found the format changes of the Modis products not necessary and basic information that is totally unneeded as it stands (P5 L29-), same with other sections (P6 L15 tiles) etc. P6 L 5-7 The use of a 4 km grid is not justified. P6 L 12 What is most confident? I found extremely confusing the paragraph P7 L3-9, this is the most important part of the methods, to clarify the type of analysis for each research question.**

We reduced the information related to the MODIS sensors in the section. Regarding the spatial resolution of the datasets, time series of monthly and annual averages presented in Figs. 3, 4 and 7 of the Discussion paper considered the entire area of the Cerrado, therefore, values corresponded to the sum of the hotspots and burned area and to the average of precipitation and VCI for the whole biome. Regarding the concern of the Referee, we have applied the Breaks For Additive Seasonal and Trend (BFAST), an additive method that decomposes a time series into seasonal, trend and noise components (Verbesselt et al., 2010), in the Cerrado 2002-2015 time series of monthly total hotspots, monthly total burned area, monthly average precipitation and monthly average VCI in order to find trends in the four time series (Fig. 4 of the revised manuscript).

Regarding the spatial results, all maps of the revised paper considered the same grid size (0.25° x 0.25°, spatial resolution of TRMM data), enabling that all results are comparable, therefore, Figs. 8, 10 and 11 of the Discussion paper were replaced by Figs. 5(a), 8 and 9 of the revised manuscript, respectively. All new figures were generated using the values of hotspots, burned area, precipitation and VCI corresponding, respectively, to the sum of monthly total hotspots, sum of monthly total burned area, the original TRMM monthly precipitation values and the monthly average VCI for each grid cell of the 0.25° x 0.25° grid over the Cerrado. This was clarified in the Data Processing section of the revised paper. Still, most confident in P6 L12 are highly reliable observations, and, therefore, the most probable pixels of being burned area.

**Results and Discussion Since there is no clear research questions, the results are hard to follow but basically the Moreira de Araujo et al 2015 paper shows the same pattern and is heavily used in the discussion, so again the question is what is the value added of this work? I found the fire density reported really high (can be interpreted as every year all km are burnt), and is confusing if it is over the 2002-2015, I think a yearly density should be calculated or a map of fire frequency for the period. The LULC and fire relationship is weakly treated. P7 L 30 first time dry season is mentioned, needs to be explained before in the intro or study area description P8 Lines3-8, consider abbreviating years, summarizing in a table, using ranges when applicable. P8 L 14-15 any other possible explanation? Human ignition? Management practices? P8 L15- 17. Is difficult to get where this point**

**originates or relates to the current study. P8L27 consider introducing these general annual trends before monthly patterns P10L2 land use is well settled, not clear the meaning P11L16-21 what about in tropical? And if soil moisture is so important, was not mentioned in the introduction.**

Considering the new analysis proposed and the comments of the Referee, we think that the value added of the work is clearer in the revised paper. Fire density reported considered the entire 2002-2015 period, and, in the revised paper, yearly density were calculated. Additionally, the relationship between LULC and fire was better explored and discussed, especially citing the work of Pivello (2011). Dry and rainy season explanation were moved to the Study Area section. Results beginning in P8 L3-8 of the Discussion paper were summarized in Table 2 of the revised paper.

In P8 L14-15 of the Discussion paper, precipitation and vegetation conditions due to the accumulated months of drought are the better explanation, however, management practices may also influence and were cited as a possible explanation in the revised paper. As a result of the new analysis, we can see that spatially there is a variation in the maximum values of hotspots and burned area and minimum precipitation and VCI and in the lag between the variables, which helps to understand why there are still high averages in October. Additionally, P8 L15-17 was rewritten. In the revised manuscript, annual trends are introduced before monthly patterns (P9 L6).

In P10 L2, land-use well settled means that land-use change is not usual in recent days, once human occupation in these areas is older and there are few natural remnants of the Cerrado, which was clarified in the revised paper. The citations in P11 L16-21 of the Discussion paper intended to present other climate controllers that have influence over fires in other vegetated areas of the globe and may also influence the occurrence of fires in the Cerrado, but were not analysed yet. Considering the comment of the Referee and once these variables were not tested for the Cerrado we removed the P11 L16-21 sentence of the Discussion paper from the revised paper.

**Conclusions. This section needs to be heavily rewritten, first two paragraphs are not conclusions but mostly repetition of results. P12 L 12 How have you established the conclusion of the Cerrado as adapted and dependent of fires? This conclusion is not clear, neither in L19-21**

The Conclusions section was entirely rewritten considering the comments of both Referees and the new results found, especially the first two paragraphs. The new Conclusions section included the following topics regarding the new analysis proposed:

- Analysing only average values may not the best approach to characterize the occurrence of fires in the Cerrado;
- Spatial analysis and its relationship with the variation of hotspots, burned area, precipitation and VCI in the Cerrado;
- Usually, there is a lag of 2 or 3 months between the minimum values of precipitation and hotspots/burned area in the Cerrado and no lag between VCI and hotspots/burned area in the biome;
- A statement regarding VCI as a good instantaneous indicator of the occurrence of fires in the Cerrado;
- More intense fires are not necessarily located in the areas where hotspots are more concentrated in the Cerrado.

**Figures. Too many, select the most relevant ones, eg. I would remove Fig**

The number of figures was reduced in the revised version of the manuscript. For example, Figs. 2, 5 and 6 of the Discussion paper were removed. Even considering the four new figures resulting from the new analysis performed, the number of figures was reduced to ten.

**The authors must be careful when talking about the lack of such specific studies. In fact, there are some studies that show the spatial and temporal patterns of fire and relation between fires and climatic variables in Cerrado. The authors should see the authors they mention (Moreira et al 2012, 2015; Pivello, 2011,) and also papers listed below: The work of Libonati et al., 2015a shows that the intra-annual variability of burnt area over the Brazilian woody savannah mostly relates to the seasonal regime of precipitation. These authors also show that there is a marked dry season from May to September, characterized by very low precipitation amounts and that, during the dry period, there is a steady displacement towards higher values of the median, lower and upper quartiles and extremes of the distributions of monthly values of burnt area. The work of Libonati et al., 2015b analyzed the results of three currently burned area products derived from MODIS data, namely AQM (INPE), MCD45A1 (NASA) and MCD64A1 (NASA). The procedure is applied to quantify the overall temporal and spatial distribution patterns of burned areas in Brazil for the period 2005 – 2010 and obtained patterns are compared for each Brazilian biome and related to the respective patterns of fire pixels derived from remote sensing. The Cerrado biome was found the one with the largest BA, followed by Caatinga and Amazônia. Estimates of BA over Brazil from AQM, MCD45 and MCD64 products for the period present a similar inter-annual variability. In addition, the work of Libonati et al., 2016 allows analyzing the overall temporal and spatial distribution patterns of BA for the last decade. The highest monthly mean amount is observed in September, followed by October, and March presents the lowest amount. The most severe year is 2007, followed by 2005 and 2010; 2006 and 2009 are the years with less area burned, followed by 2008. The spatial pattern of BA shows that the north region of Cerrado presents the highest frequency of occurrence. The spatial pattern of BA shows that the north region of Cerrado presents the highest frequency of occurrence. The intra and inter-annual variability of BA over Cerrado are closely related to the variability of precipitation but it is worth emphasizing that, despite the major role played by climate conditions, the human factor has also a prominent role on fire dynamics in this region and cannot be disregarded.**

This paper is not focused on Burned Area (BA) or how to estimate it. We used this variable as an additional dataset to spatially analyse fire patterns and their correlation with precipitation and vegetation condition over the Cerrado. Most of the articles cited by the Referee used averaged/summed values for the entire Cerrado and are focused only in BA and fire count. In the present study, we analysed, besides averaged/summed time series, datasets at pixel-based statistics, such as spatial correlation or lags. We included the references of Libonatti et al. in the Introduction section of the revised paper and when discussing results found for BA. Regarding the lack of studies, this sentence was corrected in the revised paper. The lack of studies is regarded analysing the correlation between hotspots/BA and climate-related drivers spatially, such as the results presented in Fig. 8 of the revised paper and in the new analysis performed. Pivello (2011), Moreira de Araújo et al. (2012), Moreira de Araújo and Ferreira (2015), and Libonati et al. (2015a, 2015b, and 2016) did not provide this information. In fact, all the works cited above, except to Pivello (2011), who presented an overview of the fire history in the Amazon and in the

Cerrado and described how fire regime changed in the biomes, are focused on BA, which was only one of the variables analysed and discussed in the paper. Two of the references are focused on describing an algorithm for BA detection and validating the MCD45A1 BA product (Libonati et al. (2015a) and Moreira de Araújo and Ferreira (2015), respectively). Moreira de Araújo et al. (2012) and Libonatti et al. (2015a) analysed the correlation between BA and precipitation in the Cerrado, however, not spatially, therefore, they did not show in which areas of the biome these two variables are more correlated.

Libonatti et al (2015b), which was published in the proceedings of the Brazilian Symposium on Remote Sensing, compared 3 different BA datasets (AQM, MCD45A1, and MCD64A1) and also analysed the spatial and temporal variability of BA in the Brazilian territory for the 2005-2010 period. Their work did not analyse the spatial and temporal variability in the area corresponding to the delimitation of the Cerrado, only nationally. We analysed the temporal distribution of hotspots and BA in the Cerrado considering a longer time series (2002-2015), as well as the spatial distribution of hotspots, not BA, in the Cerrado (Fig. 5(a) of the revised manuscript). Regarding Libonati et al. (2016), we found only a one-page abstract of the reference published in the proceedings of the EGU General Assembly. The results cited by the Referee were obtained from AQM product, are described in one paragraph and do not show the annual or monthly values of BA or the map showing that the highest concentration of BA is in the North of the Cerrado. Moreover, we have substantially discussed the role of human activities in the occurrence of fires in the savannas (from P11 L23 to the end of the Results and Discussion section of the Discussion paper), as well as in the citation of Pivello (2011) present in the Results and Discussion section of the revised paper (P11 L20-28).

**It is worth mentioning that the manuscript uses the same precipitation data from Libonati et al., 2015a.**

TRMM is the most used dataset in studies of precipitation conducted using remote sensing, once it provides excellent estimation of spatial and temporal patterns of precipitation considering a period of more than 15 years and is widely validated. The efficiency of TRMM data in the Brazilian territory is shown in works such as Pereira et al. (2013), cited in the Discussion paper, which justifies the choice of the dataset. In fact, Libonati et al. (2015a) was not the first paper that used TRMM dataset for analysing precipitation in the Cerrado. We can find several references that use the same product, for example, this reference from 2011 related to the environmental analysis in South America using TRMM and fire datasets (http://www.mdpi.com/2072-4292/3/10/2110), which is prior to the mentioned study. Furthermore, Moreira de Araújo et al. (2012) also used TRMM data when analysed the distribution patterns of burned area in the Brazilian biomes.

Moreover, Moreira de Araújo et al. (2012) and Libonati et al. (2015a) used an average value for the entire Cerrado and did not considered the spatial variability of precipitation, which includes distinct mechanisms. In the Cerrado, we have a substantial variation of the dry season peak, for example, the North region of the biome is mainly controlled by the Intertropical Convergence Zone and Upper Level Cyclonic Vortex disturbances, while the South region of the biome is mainly controlled by anticyclones and cold fronts. Thus, using only average values for the entire biome may not be the best approach. Therefore, we performed spatial approach with statistical analysis pixel-by-pixel, as presented in Figs. 6, 7, 8, 9 and 10 of the revised manuscript.

**Accordingly, all the results shown in the manuscript were previously reported by the above studies.**

VCI analysis was not present in any of the studies cited by the Referee, as well as the spatial correlations presented in the Discussion paper. Nevertheless, the trend analysis, spatial analysis of the month with highest incidence of hotspots and burned area, minimum amount of precipitation and minimum VCI in the Cerrado, and the lag in months between the minimum of precipitation and maximum of hotspots, minimum of precipitation and maximum of burned area, minimum of VCI and maximum of hotspots and minimum of VCI and maximum of burned area performed and presented in the revised paper are also not provided in the works cited by the Referee, highlighting the novelty of the new version of the manuscript.

**For instance, the work of Veraverbeke et al. (2014) is no about social and economic costs of fires, but on remote sensing techniques aiming fire severity assessment. In addition, the authors argue the efficiency of MCD45A1 product for mapping and understanding fire behavior and its impacts in the Cerrado. These results are not in agreement with previous works such as Roy et al (2008) and Libonati et al. (2015a), who have pointed out an under-detection of BA by the MCD45 product in savannas regions of Brazil.**

The citation of the social and economic costs of fires was corrected in the revised paper; works of Brunson and Tanaka (2011) and Stephenson et al (2013) were cited. Regarding the efficiency of the MCD45A1 product for mapping and understanding fire behavior and its impacts in the Cerrado, this sentence was rewritten, and the works of Roy et al. (2008) and Libonati et al. (2015a) were cited in the revised paper (P8 L18-24). Moreira de Araújo and Ferreira (2015) analysed the performance assessment of the BA product MCD45A1 in the Cerrado by comparing the product with BA maps derived from Landsat images and found good results, however, they analysed only BA detected during September, which is the month with highest concentration of BA in the Cerrado.

**5) Some conclusions are not based on the results: 1) 'Nevertheless, the annual variability of hotspots and precipitation and between burned area and precipitation during the 2002-2015 period is evidenced.' 2) Besides the seasonal modulation by precipitation, fire occurrence seems to respond to its interannual variability. Drier (wetter) years are associated with more (less) fires in the studied area. There is no clear evidence in the manuscript, despite visual comparison, about the statistical significance of this relation. The authors say in the conclusions that 'The methods applied are easily implemented and can be used for analyzing the occurrence and dynamics fires in different areas of the globe'. I don't think this work shows a 'method'. Instead, it is mainly devoted to a spatial and seasonal description of available variables.**

The Conclusions section was entirely rewritten considering the comments of both Referees and the new results found, especially the first two paragraphs, also improving the relationship of the variables, not only described them. The word 'method' was replaced by 'approach'. The new Conclusions section included the following topics regarding the new analysis proposed:
- Analysing only average values are not the best approach to characterize the occurrence of fires in the Cerrado;

- Spatial analysis and its relationship with the variation of hotspots, burned area, precipitation and VCI in the Cerrado;
- Usually, there is a lag of 2 or 3 months between the minimum values of precipitation and hotspots/burned area in the Cerrado and no lag between VCI and hotspots/burned area in the biome;
- A statement regarding VCI as a good indicator of the occurrence of fires in the Cerrado;
- More intense fires are not necessarily located in the areas where hotspots are more concentrated in the Cerrado.

---

## Author Response (AR3)

Dear Editor and Referee,

We have now performed the minor changes recommended by the Referee #2 in the new version of the manuscript. Answers for the all suggestions raised are presented below. We sincerely hope you find the revised manuscript suitable for publication and to hear from you soon.

1. **Fire risk models are usually based on previously (pre-fire) information about lack of precipitation, vegetation condition, humidity, temperature. So I do not agree that VCI information simultaneous with fire occurrence could help in fire risk models. In this case, I think VCI could at least be an indicator of the area affected by the fire. However, you agree that you cannot conclude that VCI minimum values are due to meteorological conditions (lack of precipitation), due to fire occurrence or due to the synergic effect of both. In such context, you do not have any evidence about it and cannot affirm that VCI is good or not for fire risk models/burned area information. So, you should remove the text (in both abstract and conclusions): 'qualifying the index as an input for fire risk models'.**

We have accepted the suggestion of the Referee. The sentence 'qualifying the index as an input for fire risk models' was replaced in the Abstract and Conclusion sections by the sentence 'qualifying VCI as an indicator of the susceptibility of vegetation to ignition' (P1L21-22 and P17L7, respectively).

2. **P16L25-27. I suggest brief discussions about the implication of the obtained trends, concerning you are using only 14 years.**

We have briefly discussed the trends found considering a time series of only 14 years in P16L29-31.

3. **Please check the citation and reference:**

**JOLLY et al., 2015**

**Joly, W. M., Cochrane, M. A., Freeborn, P. H., Holden, Z. A., Brown, T. J., Williamson, G. J., and Bowman, D. M. J. S.: 10 Climate-induced variations in global wildfire danger from 1979 to 2013, Nature Comms, 6, doi:10.1038/ncomms8537, 2015.**

Jolly et al. (2015) citation was corrected in the References section (P20L14-15).

4. **P4l26 – MCD45A1 insert reference:**

**D.P. Roy, Y. Jin, P.E. Lewis, C.O. Justice. Prototyping a global algorithm for systematic fire-affected area mapping using MODIS time series data. Remote Sens. Environ., 97 (2005), pp. 137-162, 10.1016/j.rse.2005.04.007.**

Roy et al. (2005) was cited in P4L26.

5. **P5L1314- MOD13A3 insert reference:**

**K. Didan. (2015). MOD13A3 MODIS/Terra vegetation Indices Monthly L3 Global 1km SIN Grid V006. NASA EOSDIS Land Processes DAAC. https://doi.org/10.5067/modis/mod13a3.006.**

Didan (2015) reference was inserted in P5L16.

**6. P6L2-MCD12Q1 insert reference:**

**Friedl, M.A., D. Sulla-Menashe, B. Tan, A. Schneider, N. Ramankutty, A. Sibley and X. Huang (2010), MODIS Collection 5 global land cover: Algorithm refinements and characterization of new datasets, 2001-2012, Collection 5.1 IGBP Land Cover, Boston University, Boston, MA, USA.**

Friedl et al. (2010) reference was inserted in P6L3.

**7. These references are not cited:**

**Prentice, I. C., Roos, C. I., Scott, A. C., Swetnam, T. W., van der Werf, G. R., and Pyne, S. J.: Fire in the Earth System, Science, 324, 5926, doi:10.1126/science.1163886, 2009.**

The authors presented above are part of the reference Bowman et al. (2009) (P18L14-17), not a new reference.

**Straschnoy, J. V., CRC Press, Boca Raton, United States of America,125-148, 2013.**

This is not a new reference, it is part of the Shimabukuro et al. (2013) reference (P22L32-34).

**8. Please insert e), f), g), h) in figure 2 and its caption. Also provide information about boxplot (i.e. how the median, outliers and quartiles are shown). Please also insert in each figure the respective y axis label.**

Figure 2 in the new version of the manuscript follows the suggestions of the Referee (P24).

**9. I am not sure if Moreira de Araújo et al. (2012), Libonati et al. (2015a), Libonati et al. (2015b), Moreira de Araújo and Ferreira (2015), and Libonati et al. (2016) have used the same burned area dataset (MCD45). Please specify.**

Moreira de Araújo et al. (2012) and Moreira de Araújo and Ferreira (2015) used only MCD45A1 product (specified in P8L19-22 and P8L27-28, respectively). Libonati et al. (2015a) used MCD45A1, MCD64A1 and AQM burned area datasets (specified in P8L24-26). Libonati et al. (2015b) used MCD45A1, MCD64A1 and AQM burned area datasets; the not good performance assessment in dense vegetation areas cited in P8L22-24 referred to MCD45A1 product, while the spatial pattern of burned area in the Cerrado cited in P13L12-13 was found using the AQM product. Libonati et al. (2016) used the AQM burned area dataset (specified in P13L12-13).

[revised manuscript text omitted]